# Towards Hyperparameter-free Policy Selection for Offline Reinforcement Learning

**Siyuan Zhang**
Computer Science
University of Illinois at Urbana-Champaign
siyuan3@illinois.edu

**Nan Jiang**
Computer Science
University of Illinois at Urbana-Champaign
nanjiang@illinois.edu

## Abstract

How to select between policies and value functions produced by different training algorithms in offline reinforcement learning (RL)—which is crucial for hyperparameter tuning—is an important open question. Existing approaches based on off-policy evaluation (OPE) often require additional function approximation and hence hyperparameters, creating a chicken-and-egg situation. In this paper, we design hyperparameter-free algorithms for policy selection based on BVFT [XJ21], a recent theoretical advance in value-function selection, and demonstrate their effectiveness in discrete-action benchmarks such as Atari. To address performance degradation due to poor critics in continuous-action domains, we further combine BVFT with OPE to get the best of both worlds, and obtain a hyperparameter-tuning method for $Q$-function based OPE with theoretical guarantees as a side product.

## 1 Introduction and Related Works

Learning a good policy from historical data without interactive access to the actual environment, or offline (batch) reinforcement learning (RL), is a promising approach to applying RL to real-world scenarios when high-fidelity simulators are not available [LKTF20]. Despite the fast development in the training algorithms, a burning question that remains wide open is how to tune their hyperparameters, sometimes known as the offline policy selection problem [Pai+20; YDNTS20; Fu+21].

Standard approaches reduce the problem to off-policy evaluation (OPE), which estimates the expected return of the candidate policies and choose accordingly. Unfortunately, OPE itself is a difficult problem, and standard estimators such as importance sampling suffer exponential (in horizon) variance [LMS15; JL16]. While polynomial-variance estimators exist, either using TD (e.g., Fitted-Q Evaluation, or FQE [LVY19]) or marginalized importance sampling [LLTZ18; NCDL19; UHJ20], they require additional function approximation, inducing yet another set of hyperparameters (e.g., the neural-net architecture) which need to be carefully chosen. [Pai+20] recently conclude that FQE can be effective for offline policy selection, but "an important remaining challenge is how to choose hyperparameters for FQE". (Incidentally, we are able to address this question as a side product of our approach in Section 5.) In other words, to tune hyperparameters for training we need to tune the hyperparameters for OPE, creating a chicken-and-egg situation. To this end, we want to ask:

**Can we design effective *hyperparameter-free* methods for offline policy selection?**

The question has been investigated in the theoretical literature [FS11], mostly reformulated so that we select indirectly among value functions instead of policies to trade-off directness for tractability. More precisely, [FS11] imagines that training algorithms produce candidate $Q$-functions $Q_1, Q_2, \ldots, Q_m$, which is a reasonable assumption as most offline algorithms produce value functions as a side product. The goal is to select $Q_i \approx Q^\star$—assuming one exists—so that the induced greedy policy, $\pi_{Q_i}$, is near-optimal. While $\|Q - Q^\star\|$ is only a surrogate for the performance of $\pi_Q$, the hope is that whether

$Q \approx Q^\star$ can be more easily verified from holdout data without additional function approximation, possibly by estimating the Bellman error (or residual) $\|Q - \mathcal{T}Q\|$. Unfortunately, $\|Q - \mathcal{T}Q\|$ is not amendable to statistical estimation in stochastic environments [SB18]. The naïve estimator which squares the TD error (see "1-sample BR" in Proposition 3) suffers the infamous *double-sampling bias* [Bai95], and debiasing approaches demand additional function approximation (and hence hyperparameters) [ASM08; FS11]. In prototypical real-world applications, hyperparameter-free heuristics such as picking the highest $Q$ [GGMVS20] are often used despite the lack of theoretical guarantees, which we will compare to in our experiments.

In this paper, we attack the problem based on a recent theoretical breakthrough in value-function selection: [XJ21] propose a theoretical algorithm, BVFT, which provides a workaround to the double sampling issue without requiring additional function approximation; they estimate a form of projected Bellman error as a surrogate for $\|Q - Q^\star\|$, where the function class for projection is created out of the candidate $Q$'s themselves. See Section 3 for details. Our contributions are 2-fold:

1. We design a practical implementation of BVFT based on novel theoretical observations, removing its last hyperparameter which determines a discretization resolution. We empirically demonstrate that BVFT enjoys promising performance in discrete-action benchmarks such as Atari games, sometimes using 20x less data than required by FQE-based policy selection.

2. The vanilla BVFT suffers performance degradation in continuous-action benchmarks, where the training algorithms often have an actor-critic structure and output a $(\pi, Q)$ pair where $Q$ is far away from $Q^\star$ for various reasons. To address this challenge, we propose BVFT-PE, a variant of BVFT that allows us to select among $(\pi, Q)$ pairs and pick one where $Q \approx Q^\pi$ and $\pi$ yields a high return. To further handle the issue that $Q$ from the critic is often a poor fit of $Q^\pi$, we propose to use *multiple* OPE algorithms to re-fit $Q^\pi$, and run BVFT-PE among the produced $(\pi, Q)$ pairs. While the OPE algorithms often have many hyperparameters that need to be set, BVFT-PE automatically chooses between them, leaving very few to no hyperparameters untunable. This allows us to combine the strengths of OPE and BVFT and get the best of both worlds. We also show additional results that BVFT-PE can be used for hyperparamter tuning in $Q$-function-based OPE and provide theoretical guarantees, which is of independent interest.

## 2 Preliminaries

**Markov Decision Processes (MDPs)** In RL, we often model the environment as an MDP, specified by its state space $\mathcal{S}$, action space $\mathcal{A}$, reward function $R : \mathcal{S} \times \mathcal{A} \to [0, R_{\max}]$, transition function $P : \mathcal{S} \times \mathcal{A} \to \Delta(\mathcal{S})$ ($\Delta(\cdot)$ is the probability simplex), discount factor $\gamma \in [0, 1)$, and a initial state distribution $d_0$. We assume $\mathcal{S} \times \mathcal{A}$ is finite but can be arbitrarily large. A deterministic policy $\pi : \mathcal{S} \to \mathcal{A}$ induces a random trajectory $s_0, a_0, r_0, s_1, a_1, r_1, \ldots$ where $s_0 \sim d_0$, $a_t = \pi(s_t)$, $r_t = R(s_t, a_t)$, and $s_{t+1} \sim P(\cdot|s_t, a_t)$, $\forall t$. We measure the performance of $\pi$ using $J(\pi) := \mathbb{E}[\sum_{t=0}^\infty \gamma^t r_t | \pi]$. In discounted MDPs, there always exists an optimal policy $\pi^\star$ that maximizes $J(\cdot)$ for all starting states. It is the greedy policy of the optimal $Q$-function, $Q^\star$, i.e., $\pi^\star = \pi_{Q^\star} := (s \mapsto \arg\max_a Q^\star(s, a))$. $Q^\star$ is the fixed point of Bellman optimality equation, $Q^\star = \mathcal{T}Q^\star$, where $\forall f \in \mathbb{R}^{\mathcal{S} \times \mathcal{A}}$, $(\mathcal{T}f)(s, a) := R(s, a) + \gamma \mathbb{E}_{s' \sim P(\cdot|s,a)}[\max_{a'} f(s', a')]$. A related important concept is $Q^\pi$, which tells us the expected return of $\pi$ when the trajectory starts from a specific state-action pair.

**Offline Data** In offline RL, we are given a dataset of $(s, a, r, s')$ tuples and cannot directly interact with the MDP. For the theoretical part of the paper, we assume the standard *offline sampling protocol*, that the tuples are generated i.i.d. as $(s, a) \sim \mu$, $r = R(s, a)$, $s' \sim P(\cdot|s, a)$. With a slight abuse of notation, we also use $\mathbb{E}_\mu[\cdot]$ to denote the expectation over $(s, a, r, s')$ sampled as above.

**Policy Selection/Ranking** We will use the following unified framework for policy selection throughout the paper: Suppose training algorithms (or the same algorithm with different hyperparameters) produce multiple $(\pi, Q)$ pairs, $\{(\pi_i, Q_i)\}_{i=1}^m$, and our goal is to select a policy with good performance. The relationship between $\pi$ and $Q$ can differ in different contexts: for example, when training algorithms try to fit $Q^\star$ and induce a greedy policy, we have $\pi_i = \pi_{Q_i}$, and we only need to work with $\{Q_i\}_{i=1}^m$ as they contain all the relevant information. In the case where training algorithms have an actor-critic structure, $\pi$ and $Q$ are separate quantities and need to be reasoned about together. In real applications, the next step in the pipeline is to deploy the policy in the real system for online evaluation, and since we may have the resources to test more than 1 policy, we require all algorithms to produce a *ranking* over $\{\pi_i\}_{i=1}^m$, often by sorting the policies in ascending order w.r.t. a loss.

## 2.1 Experiment Setup

To avoid interrupting the flow of intertwined theoretical reasoning and empirical evaluation in the rest of the paper, we briefly describe our experiment setup here, with details deferred to Appendix C.

**Environments and Datasets** We perform empirical evaluation on OpenAI Gym [Bro+16], Atari games [BNVB13], and Mujoco [TET12]. Taxi [Die00] is used for sanity check. We use standard offline datasets when available (RLUnplugged [Gul+21] for Atari, and D4RL [FKNTL21] for MuJoCo), and generate our own otherwise by mixing a trained expert policy with 30% chance of acting suboptimally. [Fu+21] have proposed a new benchmark for offline policy selection, which is also based on RLUnplugged/D4RL which we use and has only become available very recently. Also this benchmark (and [VLJY19]) focuses on OPE and policy selection without value functions, which does not exactly fit our purposes. We leave the evaluation on their benchmarks to future work.

**Training Algorithms** We use several different training algorithms to generate the candidate models, including offline algorithms such as BCQ [FCGP19] and CQL [KZTL20]. To test the robustness of the policy-selection methods w.r.t. how the candidate policies are generated, we also use algorithms that learn from online interactions such as DQN [Mni+15] in some domains, though policy selection is always performed using separate offline datasets. This scenario is also of interest in its own right, as one can imagine training on (possibly imperfect) simulators via online algorithms and using limited realworld offline data for policy selection. For each algorithm, we consider different neural architectures, learning rates, and learning steps as hyperparameters to produce multiple candidate policies (and value functions) for selection; see Table 1 in Appendix C for details.

**Performance Metrics** In each experiment, we run different policy-selection methods and evaluate the produced policy rankings using the following two metrics adapted from [YDNTS20]:

***Top-$k$ normalized regret*** We take the best policy within the top-$k$ recommended by the ranking, and calculate its gap compared to the best policy among all candidates. This metric reflects our regret if we were to online-evaluate the top-$k$ policies in the ranking and identify the best among them. To make the value more interpretable, we normalize the regret by the gap between the best and the worst candidate policies.
***Top-$k$ precision*** We take the $k$ best policies in terms of their groudtruth values, and return the proportion of them appearing in the top-$k$ policies in the ranking.

This process is repeated for 200 or 300 runs, with randomness coming from re-sampling a subset of the dataset for policy selection (usually of size $50,000$; FQE needs much more data and we do not run it on random subsets) and sampling $m = 10$ or $15$ policies from all candidates for comparison. All figures report the mean with error bars of twice the standard errors, i.e., $95\%$ confidence intervals.

## 3 Background: Batch Value-Function Tournament (BVFT)

We briefly introduce the theoretical basis of our approach. As mentioned in Section 1, the Bellman error $\|Q - \mathcal{T}Q\|$ is an appealing quantity because $Q = Q^\star \Leftrightarrow \|Q - \mathcal{T}Q\|_\infty = 0$, but it is not amendable to statistical estimation in stochastic environments due to the double-sampling bias [Bai95; ASM08; FS11]. Given this caveat, a closely related quantity has been extensively studied in the literature: given function class $\mathcal{G} \subset [0, \frac{R_{\max}}{1-\gamma}]^{\mathcal{S} \times \mathcal{A}}$, the (mean-squared) *projected Bellman error* of $Q$ w.r.t. $\mathcal{G}$ is defined as

$$\|Q - \mathcal{T}_\mathcal{G} Q\|_{2,\mu}^2, \text{ where } \mathcal{T}_\mathcal{G} Q := \arg\min_{g \in \mathcal{G}} \mathbb{E}_\mu[(g(s,a) - r - \gamma \max_{a'} Q(s',a'))^2]. \quad (1)$$

Here $\|\cdot\|_{2,\mu}^2 = \mathbb{E}_\mu[(\cdot)^2]$, and the dependence of $\mathcal{T}_\mathcal{G}$ on $\mu$ is suppressed for readability. This quantity can be straightforwardly estimated from data when $\mathcal{G}$ has bounded statistical complexity; we just need to replace all $\mathbb{E}_\mu[\cdot]$ with their finite-sample approximation. The property of the projected Bellman error largely depends on the choice of $\mathcal{G}$, which needs to satisfy certain conditions for $\|Q - \mathcal{T}_\mathcal{G} Q\|$ to be a good surrogate for $\|Q - Q^\star\|$. The seminal work of [Gor95] has provided the following sufficient condition: (see [XJ21, Proposition 4] for a formal proof)

**Proposition 1.** *If (1) $\mathcal{G}$ is a piecewise-constant class, and (2) $Q^\star \in \mathcal{G}$, then $\mathcal{T}_\mathcal{G}$ is $\gamma$-contraction under $\|\cdot\|_\infty$, and $Q = Q^\star \Leftrightarrow \|Q - \mathcal{T}_\mathcal{G} Q\|_{2,\mu} = 0$ if $\mu$ is fully supported on $\mathcal{S} \times \mathcal{A}$.*

Being piecewise constant means that there exists a partitioning of $\mathcal{S} \times \mathcal{A}$, and $\mathcal{G}$ consists of all members of $[0, \frac{R_{\max}}{1-\gamma}]^{\mathcal{S} \times \mathcal{A}}$ that remain constant within each partition. A partitioning whose induced $\mathcal{G}$ satisfies (1) and (2) is also closely related to $Q^\star$-preserving state abstractions [LWL06].

The problem is that it is very difficult to find $\mathcal{G}$ that satisfies the above 2 criteria,[1] and it will just be another set of hyperparameters to tune if we leave the design of $\mathcal{G}$ to the user. [XJ21] offers a resolution in the context of selecting $Q^\star$ from $\{Q_i\}_{i=1}^m$: consider the base case of $m = 2$ where we only need to select between $Q_1$ and $Q_2$. If one of them is $Q^\star$ (which can be relaxed) but we do not know which, we can still create $\mathcal{G}_{1,2}$ that satisfies both criteria of Proposition 1 as the minimal piecewise-constant class such that $\{Q_1, Q_2\} \subset \mathcal{G}_{1,2}$. If the output of each $Q_i$ takes at most $N$ possible values, $\mathcal{G}_{1,2}$ will be induced by partitioning $\mathcal{S} \times \mathcal{A}$ into at most $N^2$ regions; see Figure 1L.[2] [XJ21] further shows that this idea extends to arbitrary $m$ via pairwise comparison ("tournament"): The validity of the procedure can be justified by the following simplified result:[3]

**Proposition 2** (Simplification of [XJ21]; see Appendix B.1 for a proof sketch). *If $Q^\star \in \{Q_i\}_{i=1}^m$ and $\mu$ is fully supported on $\mathcal{S} \times \mathcal{A}$, then $Q_i = Q^\star \Leftrightarrow Q_i$ having 0 BVFT-loss, where*

$$\texttt{BVFT-loss}(Q_i\,;\{Q_j\}_{j=1}^m) := \max_j \|Q_i - \mathcal{T}_{\mathcal{G}_{i,j}} Q_i\|_{2,\mu}. \tag{2}$$

For each $Q_i$, BVFT calculates its projected Bellman error w.r.t. $\mathcal{G}_{i,j}$—where $\mathcal{G}_{i,j}$ is created just like $\mathcal{G}_{1,2}$ but using $Q_i$ itself and every other $Q_j$—and scores $Q_i$ using the worst-case projected error. See [XJ21] for the complete theory that accounts for $Q^\star \notin \{Q_i\}_{i=1}^m$ and finite-sample effects. For readability we will stick to the above simplified reasoning.

**Computation** The empirical version of BVFT-loss can be computed exactly in closed form. The central step is the calculation of $\mathcal{T}_{\mathcal{G}} Q$. Recall that $\mathcal{G}$ is piecewise constant and induced by a $\mathcal{S} \times \mathcal{A}$ partitioning. For any $(\tilde{s}, \tilde{a})$, $(\mathcal{T}_{\mathcal{G}} Q)(\tilde{s}, \tilde{a})$ is simply the average of $r + \gamma \max_{a'} Q(s', a')$ over all data points $(s, a, r, s')$ where $(s, a)$ falls in the same partition as $(\tilde{s}, \tilde{a})$. Implemented with running averages, the computational complexity is $O(m^2 n)$ for $m$ candidate functions and $n$ data points in addition to $(|\mathcal{A}| + 1)mn$ candidate-function evaluations (i.e., caching $Q_i(s, a)$ and $\max_{a'} Q_i(s', a')$).

## 4    BVFT **with Automatic Resolution Selection**

Despite the appealing theoretical properties, it is unclear if BVFT can be converted into a practical algorithm. The sample complexity of estimating BVFT-loss depends on the complexity of $\mathcal{G}_{i,j}$, which is controlled by $N^2$ where $N$ is the number of possible values in $[0, \frac{R_{\max}}{1-\gamma}]$ each $Q_i$ can take. To handle the issue that $N$ can be very large or even infinite, [XJ21] discretizes the output of $\{Q_i\}_{i=1}^m$ up to $\epsilon_{\text{dct}}$ error before producing $\{\mathcal{G}_{i,j}\}$, where $\epsilon_{\text{dct}}$ needs to be carefully chosen to trade-off between discretization errors and the complexity of $\mathcal{G}$ (which is now $O(1/\epsilon_{\text{dct}}^2)$). The theoretical value of $\epsilon_{\text{dct}}$ (as in [XJ21, Theorem 2]) not only relies on unknown properties of the MDP and the data, but is also very small which makes BVFT-loss expensive to estimate. Indeed, the previous theory predicts that BVFT-loss becomes an unstable statistic when $\epsilon_{\text{dct}} \to 0$ due to the unbounded complexity of $\mathcal{G}_{i,j}$.

To resolve this issue, we draw an interesting connection between BVFT-loss and the naïve "1-sample" estimator for $\|Q - \mathcal{T}Q\|$, which reveals the unexpected behavior of BVFT-loss in the regime of $\epsilon_{\text{dct}} \to 0$ that differs from the previous theoretical predictions:

**Proposition 3.** *Consider $\mathbb{E}_\mu[(Q(s, a) - r - \gamma \max_{a'} Q(s', a'))^2]$, the naïve and biased estimator for $\|Q - \mathcal{T}Q\|_{2,\mu}^2$ which we call "1-sample BR (Bellman residual)". If each candidate $Q_i$ never*

---

[1]The ideal $\mathcal{G}$ can be created similarly to Figure 1 according to knowledge of $Q^\star$. In Appendix D we show that our approach closely tracks the skyline of using the ideal $\mathcal{G}$ in a tabular domain where we can compute $Q^\star$.

[2]A common misconception is that BVFT only works under the "assumption" that $Q^\star$ is piecewise constant under certain given partition or metric over $\mathcal{S} \times \mathcal{A}$; this is not the case. The concept of piecewise-constant functions is used as an *internal mechanism* in BVFT and is not an assumption in any way. The existence of $\mathcal{G}_{1,2}$ is *unconditional* and does not rely on any structural properties of the underlying MDP. Constructing the partition (and hence $\mathcal{G}_{1,2}$) only requires $Q_1$ and $Q_2$, and does not require any additional prior knowledge (including but not limited to a pre-defined metric or partition over $\mathcal{S} \times \mathcal{A}$).

[3]Among all the $(i, j)$ pairs considered in BVFT-loss, many of them will not satisfy $Q^\star \in \{Q_i, Q_j\}$ and thus the previous reasoning for the $m = 2$ case and Proposition 1 do not apply. Despite this, these $(i, j)$ pairs do not affect the validity of the BVFT-loss; see [XJ21] for a detailed explanation.

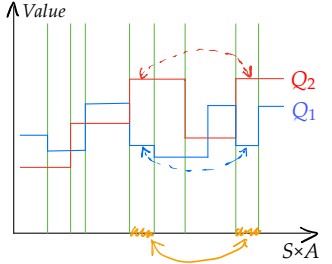 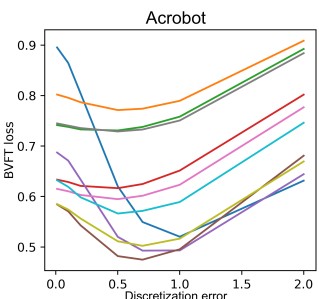

Figure 1: **Left:** Visualization of the partitioning that induces $\mathcal{G}_{1,2}$. The 2 subsets of $\mathcal{S} \times \mathcal{A}$ marked orange belong to the same partition despite being separated from each other, because $Q_1$ and $Q_2$ are constant across them. Since $\mathcal{S} \times \mathcal{A}$ is partitioned according to $Q$'s *output*, the number of partitions is *independent* of $|\mathcal{S} \times \mathcal{A}|$, allowing BVFT to scale to arbitrarily complex state-action spaces with limited data. **Right:** BVFT-loss vs. discretization error $\epsilon_{\mathrm{dct}}$ of 10 candidate $Q$'s of a typical run in Acrobot, all having the U-shape predicted by our theoretical reasoning.

predicts the same value for any two $(s, a)$ pairs seen in the dataset, then with $\epsilon_{dct} = 0$, the empirical version of BVFT-loss$(Q_i; \{Q_j\}_{j=1}^m)$ and 1-sample BR of $Q_i$ coincide (up to squaring).

*Proof.* Given a dataset $D$ consisting of $(s, a, r, s')$ tuples, the empirical version of 1-sample BR is $\frac{1}{|D|} \sum_{(s,a,r,s') \in D} (Q(s,a) - r - \gamma \max_{a'} Q(s', a'))^2$, and that of BVFT-loss squared is $\max_j \frac{1}{|D|} \sum_{(s,a,r,s') \in D} (Q(s,a) - (\widehat{\mathcal{T}}_{\mathcal{G}_{i,j}} Q)(s, a))^2$, where $\widehat{\mathcal{T}}_{\mathcal{G}_{i,j}}$ is the empirical version of $\mathcal{T}_{\mathcal{G}_{i,j}}$ based on the same dataset. It suffices to show that for any data point $(s, a, r, s')$, $(\widehat{\mathcal{T}}_{\mathcal{G}_{i,j}} Q)(s, a) = r + \gamma \max_{a'} Q_i(s', a')$, which follows immediately from $\epsilon_{\mathrm{dct}} = 0$ and $Q_i$ never predicting the exact same value twice, since $\mathcal{G}_{i,j}$ does not provide any aggregation over data points in this case and $\widehat{\mathcal{T}}_{\mathcal{G}_{i,j}}$ coincides with the 1-sample Bellman update (c.f. the paragraph on computation in Section 3).  □

This result implies that, 1-sample BR, which is a reasonable objective and coincides with $\|Q - \mathcal{T}Q\|$ in deterministic environments, provides a safeguard to BVFT-loss when $\epsilon_{\mathrm{dct}}$ is too small. Since the double-sampling bias of 1-sample BR is positive [Bai95], we expect BVFT-loss to gradually decrease as $\epsilon_{\mathrm{dct}}$ increases, hit a minimum, and increase again due to discretization errors when $\epsilon_{\mathrm{dct}}$ becomes too large.[4] Indeed, this is precisely what we observe empirically; see Figure 1R.

Based on this novel observation, we propose to search for a grid of discretization errors in BVFT and pick the resolution that minimizes the loss (Eq.(2)); see pseudocode in Appendix A. In the experiments we will always use this rule to automatically select the discretization resolution for BVFT and its variants. The remaining questions can only be answered empirically: Does BVFT ever exhibit more interesting behavior than 1-sample BR (especially given their intimate relationship), is our resolution selection rule a good one, and how does BVFT compare to other baselines?

**Empirical Evaluation**  To answer these questions, we empirically compare BVFT, 1-sample BR, and another simple hyperparameter-free heuristic, AvgQ [GGMVS20], which simply ranks $\{Q_i\}_{i=1}^m$ based on their average value on the data. "Random" ranks the policies in a completely random manner. BVFT and 1-sample BR rank $\{Q_i\}_{i=1}^m$ in ascending order of their loss functions, respectively. The experiments are done on 5 Atari games (Figure 2) and 4 Gym control problems (Figure 3; action space is made discrete in Pendulum).

• *Does BVFT ever exhibit more interesting behavior than 1-sample BR?* Perhaps surprisingly, BVFT deviates significantly from the behavior of 1-sample BR, and almost always outperforms the latter by a wide margin. This is particularly interesting in CartPole and Pendulum, where the deterministic dynamics make 1-sample BR an **unbiased** estimate of $\|Q - \mathcal{T}Q\|$, and we expected it to perform well. Contrary to our expectation, 1-sample BR performs poorly even in these domains, where BVFT often performs much better. In fact, we observe similar phenomenon in a version of Taxi where we can compute $\|Q - \mathcal{T}Q\|$ as a skyline; see Appendix D.

---

[4]This can be violated in some extreme cases; see Appendix A for details.

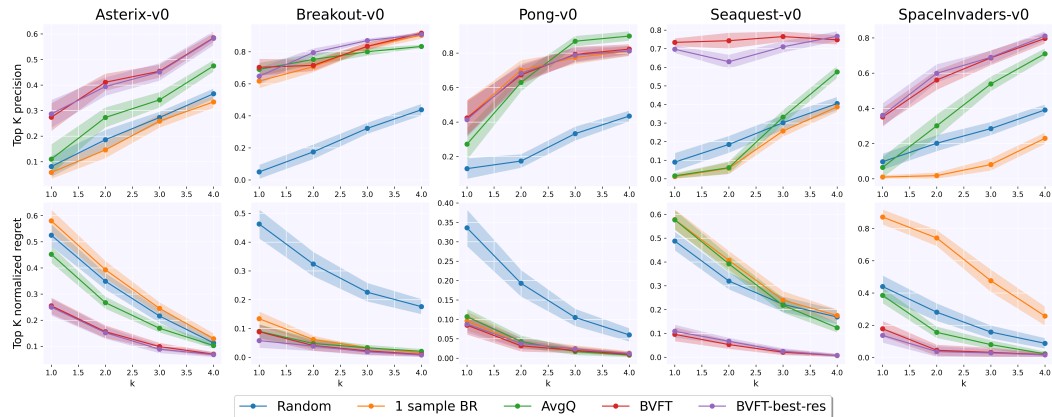

Figure 2: Top-$k$ metrics of policy rankings vs. $k$ in Atari. Row 1 shows top-$k$ precision (the higher the better), and Row 2 shows top-$k$ regret (the lower the better). Training algorithms are BCQ with different hyperparameters. The dataset for policy selection has 50,000 transition, which is an order of magnitude less than needed by FQE in Atari (see FQE in Enduro [VLJY19], as well as Figure 4).

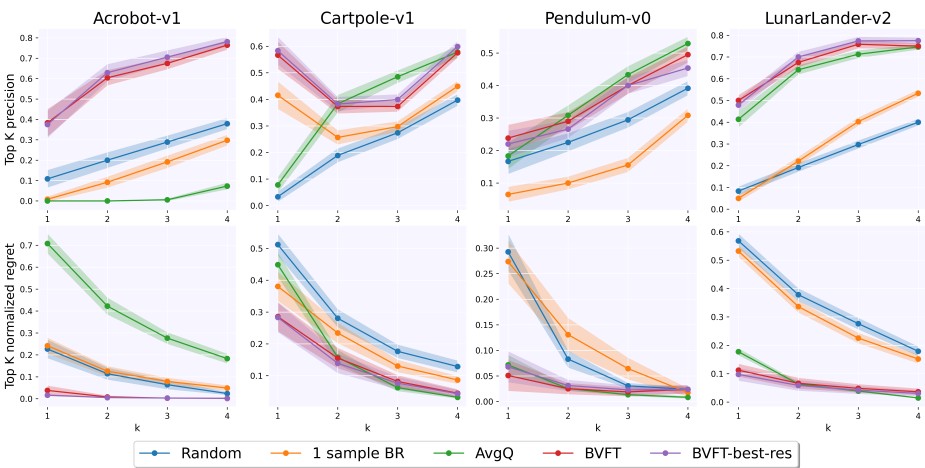

Figure 3: Results in Gym control problems. The dataset for policy selection has 50,000 transitions.

• *Is our resolution selection rule effective?* To examine the effectiveness of our resolution selection rule, we compare to a skyline of BVFT itself ("BVFT-best-res"), where the best fixed resolution is chosen based on average statistics in the hindsight. As we can see, BVFT with automatic resolution selection closely tracks the performance of BVFT-best-res—to the extent that they are mostly indistinguishable—indicating that our resolution selection rule is near-optimal.

• *Comparison to AvgQ.* The simple heuristic of picking $Q$ with the highest predicted value—which is sometimes used in prototypical applications [GGMVS20]—can be surprisingly effective sometimes, such as in Pendulum, LunarLander, and Pong. BVFT performs equally well in these domains. Moreover, as results from other domains reveal, AvgQ is very unstable and can fail catastrophically, such as in Acrobot and Seaquest, and is much less robust compared to BVFT.

One may wonder if AvgQ works better with *pessimistic* training algorithms: if every $Q_i$ is an underestimation of the true return, then maximizing $Q$ will be a well-justified heuristic. However, such pessimism is not always guaranteed especially when the function approximation is misspecified. In Appendix D we show additional experiments in Atari with a pessimistic algorithm CQL, and the results are qualitatively similar to Figure 2.

**Comparison to OPE** Despite the lack of known method for tuning OPE's hyperparameters except for "cheating" in the simulator using online roll-outs (in Section 5 we will combine BVFT with OPE to address this issue), which makes it very difficult to have fair comparisons with OPE-based

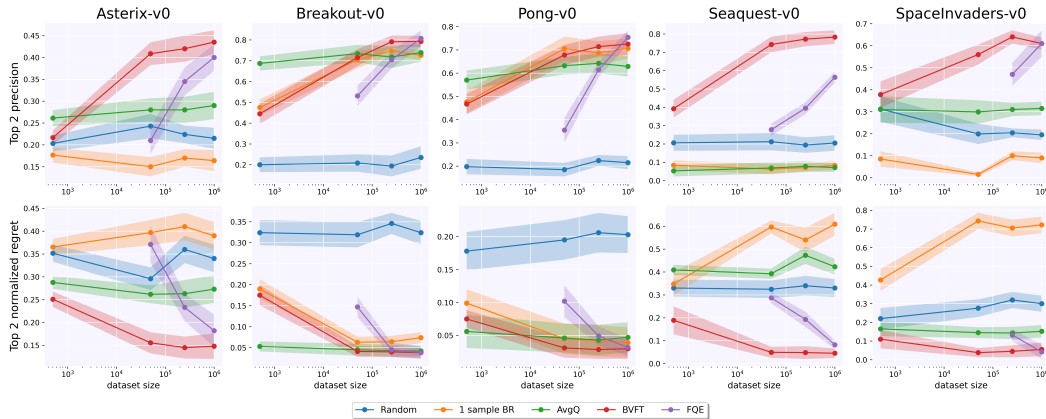

Figure 4: Comparison to FQE on Atari games. The figures show top-2 metrics vs. sample sizes. We did not test FQE on small sample sizes since the trend of performance degradation is clear.

policy-selection methods, it is still instructive to have a rough sense of how our method compares to OPE. In this and the next sections, we choose Fitted Q-Evaluation (FQE) as a representative OPE algorithm.[5] Figure 4 shows the ranking metrics vs. sample size in 5 Atari games. We tune the neural architecture for FQE by choosing the training architecture that produces the *best* policy in Asterix.[6] While FQE has equally good performance compared to BVFT in the large-sample regime ($10^6$ transitions, which is typically required for FQE in Atari), the performance degradation is severe when sample size decreases. In comparison, BVFT is much more sample-efficient and provides almost the same level of performance with about 20x less samples, which is also what we used in Figure 2.

**Additional Results**  In Appendix D we examine the sensitivity of different methods to the dataset size and exploratoriness. BVFT remains advantageous compared to the baselines, is insensitive to data exploratoriness, and enjoys improved performance when more data becomes available unlike other hyperparmeter-free baselines.

## 5   Fighting Poor Value-Function Estimates with BVFT + OPE

The experiments so far are on discrete-action domains, and the candidate policies are always greedy w.r.t. the $Q$-function. However, in continuous-action domains, actor-critic-type algorithms are often used, where a critic $Q$ is trained to evaluate a parameterized policy $\pi$, and the actor $\pi$ is in turn improved based on $Q$. The trained $(\pi, Q)$ pair in general does not satisfy $\pi = \pi_Q$, so it seems unwise to ignore $\pi$ and only focus on $Q$ in policy selection. On a related note, $\arg\max_{a'}$ is often expensive if not infeasible to calculate in continuous action spaces, so we need to make changes to BVFT anyway.

More importantly, the success of BVFT relies on the existence of a reasonable approximation of $Q^\star$ among the candidate functions, which does not always hold especially for actor-critic algorithms. Indeed, in the Mujoco experiments shown in Figure 5, we observe poor performance of all hyperparameter-free methods, including variants of BVFT for continuous-action domains we develop later, and no method can consistently outperform the trivial baseline of random ranking. We address this issue of poor critics in the rest of this section. The issue is quite complicated and has many contributing factors—as we will explain subsequently—requiring us to take a multi-step approach to address one factor at a time.

**Step 1:** BVFT-PE **and** BVFT-PE-Q **for Joint Selection of** $(\pi, Q)$  We first address the issue that $\arg\max_{a'}$ may be infeasible to calculate in continuous-action domains, and that both $\pi$ and $Q$ need to be taken into consideration for actor-critic algorithms. To address this issue, we propose a variant

---

[5]Despite recent exciting developments in marginalized importance sampling (MIS), FQE often shows strong empirical performance thanks to its simplicity [VLJY19; Pai+20; Fu+21], whereas MIS is not as off-the-shelf due to its difficult optimization, so we defer the comparison as well as combining BVFT with MIS to future work.

[6]The recent OPE benchmarks [Pai+20; Fu+21] do not include Atari, so we choose our own hyperparameters.

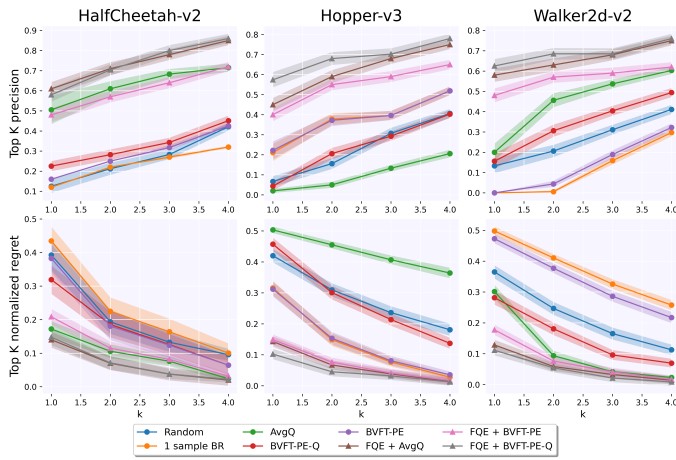

Figure 5: Top-$k$ metrics vs. $k$ across 3 Mujoco domains with continuous actions. All methods use a dataset of size $50,000$, except that the FQE component of any method uses $10^6$.

of BVFT, called BVFT-PE, with the following loss function:

$$\texttt{BVFT-PE-loss}((\pi_i, Q_i) \, ; \{(\pi_j, Q_j)\}_{j=1}^m) := \max_j \|Q_i - \mathcal{T}_{\mathcal{G}_{i,j}}^{\pi_i} Q_i\|_{2,\mu}, \tag{3}$$

where $\mathcal{G}_{i,j}$ is exactly the same as in BVFT, and $\mathcal{T}_{\mathcal{G}_{i,j}}^{\pi_i}$ is the same as Eq.(1), except that $\max_{a'} Q(s', a')$ should be replaced by $Q(s', \pi(s'))$. (In actor-critic algorithms, $\pi$ is often a stochastic policy, in which case the term means $\mathbb{E}_{a' \sim \pi(\cdot|s')}[Q(s', a')]$.) This loss function naturally generalizes the BVFT-loss: when each $\pi_i$ is the greedy policy of $Q_i$, we immediately have $\mathcal{T}_{\mathcal{G}_{i,j}}^{\pi_i} Q_i = \mathcal{T}_{\mathcal{G}_{i,j}} Q_i$, thus recovering BVFT-loss as a special case.

A closer inspection of Eq.(3) reveals an interesting fact: BVFT-PE-loss$((\pi_i, Q_i)) = 0$ as long as $Q_i = Q^{\pi_i}$, so BVFT-PE-loss is really a loss function for *policy evaluation*, hence the name BVFT-PE. (We will actually provide the theoretical guarantees of BVFT-PE for policy evaluation at the end of this section; see Theorem 4.) While the loss recovers BVFT-loss when $\pi = \pi_Q$, more generally there can exist the degenerate cases of $Q = Q^\pi$ but $\pi$ itself being a poor policy, where BVFT-PE-loss$((\pi, Q))$ is still $0$. We address this issue by subtracting a $Q$ term from the BVFT-PE-loss: BVFT-PE-Q $:=$ BVFT-PE-loss $- \lambda \mathbb{E}_\mu[Q]$, and the structure of this loss resembles a telescoping identity for $J(\pi)$ commonly used in the OPE literature; see Appendix A.1 for details.

**Step 2: Re-fitting $Q$ with OPE** The above derivations address some of the basic theoretical issues, but still implicitly assume that at least one good $\pi_i$ is paired with a $Q_i \approx Q^{\pi_i}$. In actor-critic algorithms, however, we often observe that the actor $\pi$ converges way before the critic $Q$ does, leaving us with a poorly estimated $Q$. Indeed, we have already seen in Figure 5 that BVFT-PE and BVFT-PE-Q are still not effective when applied to the candidate $(\pi, Q)$ pairs.

A natural solution to the problem is to use OPE algorithms to refit $Q^\pi$ to replace the critic $Q$, in order to provide a more accurate value function. We implement this using FQE as the OPE algorithm, though in principle we can use any OPE algorithm that provides an estimate of $Q^\pi$, including kernel loss [FLL19], MQL [UHJ20], or even a model-based method through planning. Figure 5 shows that BVFT-PE and BVFT-PE-Q enjoy strong empirical performance. However, the baseline of simply ranking the policies according to FQE itself ("FQE+AvgQ") is equally effective, which suggests that most of the performance gains should be attributed to FQE. That said, even if one's conclusion is that "OPE-based policy selection is more superior in continuous-action domains", we are just all the way back to where we started in Section 1:

*How can we tune the hyperparameters for OPE itself?*

**Step 3: Re-fitting $Q$ with *Multiple* OPE Algorithms** Our last idea is retrospectively simple, yet allows us to combine the strengths of BVFT and OPE and get the best of both worlds. Suppose we have $L$ OPE algorithms which we wish to select from. We will simply run each OPE algorithm on

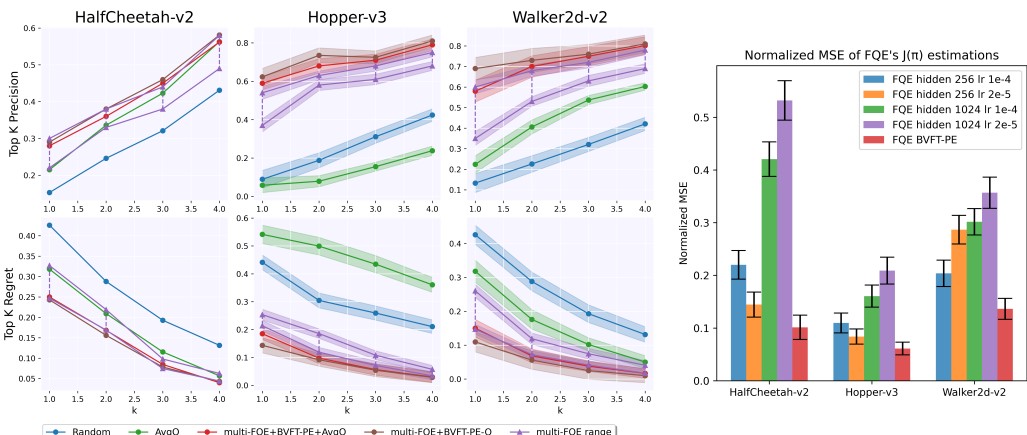

Figure 6: **Left:** Policy selection with multiple FQE instances in Mujoco. To avoid cluttering, we show the range of FQE performance across different hyperparameters (with upper and lower bounds connected by vertical dashed lines), and have removed error bars from HalfCheetah for better visibility (their widths are similar to the other two figures). The curves are cluttered in HalfCheetah and both our strategies perform similarly to the upper bound. **Right:** OPE accuracy for using `BVFT-PE` for hyperparameter tuning in FQE, where $J(\pi)$ is approximated by $\mathbb{E}_{s \sim d_0}[Q(s, \pi(s))]$.

each policy, producing $m \times L$ candidate $(\pi, Q)$ pairs in total, in the form of $\{(\pi_i, Q_i^l)\}$, where $Q_i^l$ is the estimation of $Q^{\pi_i}$ by the $l$-th OPE algorithm. There are two strategies we can proceed with:

**Strategy 1:** Run `BVFT-PE-Q` on all $m \times L$ pairs of $(\pi, Q)$. Rank the $m$ policies according to their highest position in the ranking produced by `BVFT-PE-Q`.

**Strategy 2:** Within each $\pi_i$, use `BVFT-PE` to select $Q_i^{l_i^\star} \approx Q^\pi$, then rank the policies based on the predictions of $Q_i^{l_i^\star}$ (e.g., using AvgQ).

**Applicability to Arbitrary Training Algorithms** We motivated `BVFT-PE` and `BVFT-PE-Q` by actor-critic training and assumed that training algorithms must produce $(\pi, Q)$ pairs. However, the approaches proposed above can be readily applied to any training algorithms: we discard the $Q$ from training in Step 2, and therefore the training algorithms do not need to produce value functions in the first place.

**Empirical Evaluation** We use these strategies in the same experiment setting as Figure 5, with multiple instances of FQE using different hyperparameters (see legend of Figure 6R).[7] The results are shown in Figure 6L, where both strategies perform similarly to or better than the best FQE instance. We emphasize that this is highly nontrivial, because `BVFT-PE` and `BVFT-PE-Q` do not observe the *identities* of the FQE algorithms across different policies and runs (in fact, each FQE algorithm is represented as merely $2mn$ numbers), so matching the best FQE performance is strong evidence for BVFT algorithms' model selection capabilities. Between the two BVFT strategies, Strategy 1 (using `BVFT-PE-Q`) slightly outperforms Strategy 2, but comes with an additional hyperparameter $\lambda$; we tuned it on Hopper and use the same constant in all experiments. In comparison, Strategy 2 is only slightly worse and does not have its own hyperparameters, which is an advantage.

**Hyperparameter Tuning in OPE** In Strategy 2, we are basically using OPE as the policy-selection algorithm, and `BVFT-PE` as a subroutine for hyperparameter tuning for OPE itself. Since OPE is an important component of the offline RL pipeline in its own right, our procedure for tuning OPE algorithms using `BVFT-PE` is also of independent interest. To this end, we conduct additional experiments in Mujoco to test the OPE accuracy of FQE with different hyperparameters. As Figure 6R shows, FQE with hyperparameters tuned by `BVFT-PE` consistently outperforms the *best fixed* set of hyperparameters across all 3 Mujoco domains. This implies that `BVFT-PE` can select the best hyperparameters for each policy individually, which is an appealing property. We also supplement the

---

[7]Here the training algorithms are BCQ. In Appendix D we reproduce Figure 6L with CQL as the training algorithms; see Figure 8. The results are qualitatively the same.

empirical results with a theoretical guarantee for selecting $Q^\pi$ out of candidate functions for a fixed $\pi$, which follows from similar proof techniques as [XJ21, Theorem 2]; see Appendix B for the proof.

**Theorem 4.** *Let $C$ be the same as in [XJ21, Theorem 2], which characterizes the exploratoriness of the data distribution. Consider any policy $\pi$ and candidate $Q$-functions, $\{Q^l\}_{l=1}^L$, with $Q^l \in [0, \frac{R_{\max}}{1-\gamma}]$. Let $(\pi, \hat{Q})$ be the pair that minimizes the empirical version of BVFT-PE-loss applied to $\{(\pi, Q^l)\}_{l=1}^L$ with $\epsilon_{dct} = \frac{\epsilon R_{\max}}{8\sqrt{C}}$. Then, using a dataset of size $\tilde{O}\left(\frac{C^2 \ln(L/\delta)}{\epsilon^4(1-\gamma)^4}\right)$ where $\tilde{O}$ suppresses logarithmic terms, w.p. $\geq 1-\delta$, $\sup_{\nu:\|\nu/\mu\|_\infty \leq C} \|\hat{Q} - Q^\pi\|_{2,\nu} \leq \epsilon \cdot \frac{R_{\max}}{1-\gamma} + \frac{(2+4\sqrt{C})\min_l \|Q^l - Q^\pi\|_\infty}{1-\gamma}$.*

As the theorem implies, when one of $\{Q^l\}_{l=1}^L$ is a good approximation of $Q^\pi$ (i.e., $\min_l \|Q^l - Q^\pi\|_\infty$ is small), BVFT-PE is able to identify $\hat{Q} \approx Q^\pi$ with a polynomial sample complexity.

## 6 Discussion and Conclusion

We present BVFT and its variants based on recent theoretical advances [XJ21], which are (nearly) hyperparameter-free algorithms for policy selection in offline RL and empirically effective in discrete-action benchmarks. When combined with OPE algorithms such as FQE, variants of BVFT are also competitive in continuous-action benchmarks, and such a combination addresses the weaknesses of BVFT (relying on the existence of good value functions among the candidates) and those of OPE (having untunable hyperparameters) and gets the best of both worlds.

We conclude the paper with discussions of the limitations of our approach and open questions:

**Sample Efficiency** Section 5 uses OPE to fit $Q^\pi$, which can be data intensive. A plausible solution is to re-use the training data for OPE, as suggested by [Pai+20]. However, data reuse can cause serious issues in realworld domains where the amount of training data taken for granted in deep RL is not available. How to improve the sample efficiency of BVFT + OPE is an important open question.

**Computational Complexity** The $O(m^2)$ complexity of BVFT can be prohibitive if we wish to compare hundreds of models. A plausible solution is divide-and-conquer, i.e., eliminating bad models by running BVFT in smaller groups. It will be interesting to see if this compromises performance.

**Applicability** It is important to understand what type of domains BVFT is particularly suited for. Despite having provided partial answers (e.g., the discussion of discrete actions vs. continuous actions), we still need a more thorough answer based on comprehensive evaluation across more diverse domains and comparison to a wider range of baselines, which is beyond the scope of this paper since we focus on algorithm development and some of our contributions are orthogonal to existing approaches (e.g., BVFT + OPE). We look forward to investigating this question empirically in the future with the help of the public benchmarks that have become available recently [Fu+21].

**Data with Insufficient Coverage** While results in Appendix D show that BVFT is insensitive to moderate changes in data exploratoriness, we will likely need to incorporate some form of pessimism–which is shown to be important for training [LSAB19; LSAB20; KRNJ20; JYW20; RZMJR21; XCJMA21]—when the data coverage is seriously lacking. This can be, however, quite challenging, as BVFT uses a dynamic function space for projection ($\mathcal{G}_{i,j}$) that varies from candidate to candidate, and states considered covered due to generalization effects under one function space may be considered lacking data under a different one. How to resolve this issue and design a pessimistic version of BVFT is an interesting question.

## Acknowledgments and Disclosure of Funding

Nan Jiang acknowledges funding support from the ARL Cooperative Agreement W911NF-17-2-0196, NSF IIS-2112471, and Adobe Data Science Research Award.

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
