# A Algorithm Details

---

**Algorithm 1:** BVFT with Automated Resolution Selection

---

**Input:** Dataset $D$, candidate functions $\{Q_i\}_{i=1}^m$, discretization parameter set $\mathcal{R} = \{\epsilon_k\}$

**for** $\epsilon_k \in \mathcal{R}$ **do**

    **for** $i = 1$ *to* $m$ **do**

        $\overline{Q_i} \leftarrow$ discretize the output of $Q_i$ with resolution $\epsilon_k$.

    **end**

    **for** $i = 1$ *to* $m$ **do**

        **for** $j = 1$ *to* $m$ **do**

            Define $\phi_{i,j}$ as a partitioning of $\mathcal{S} \times \mathcal{A}$ s.t. $\phi(s,a) = \phi(s',a')$ iff $\overline{Q_i}(s,a) = \overline{Q_i}(s',a')$ and $\overline{Q_j}(s,a) = \overline{Q_j}(s',a')$;

            Let $\mathcal{G}_{i,j}$ be the piecewise constant function class induced by $\phi_{i,j}$;

            $\hat{\mathcal{T}}_{\mathcal{G}_{i,j}} Q := \arg\min_{g \in \mathcal{G}_{i,j}} \frac{1}{|D|} \sum_{(s,a,r,s') \in D} [(g(s,a) - r - \gamma \max_{a'} Q(s',a'))^2]$;

            $\mathcal{E}_{\epsilon_k}(Q_i; Q_j) \leftarrow \|Q_i - \hat{\mathcal{T}}_{\mathcal{G}_{i,j}} Q_i\|_{2,D}$    # $\|f\|_{2,D}^2 := \frac{1}{|D|} \sum_{(s,a) \in D} f(s,a)^2$ ;

        **end**

        $\mathcal{E}_{\epsilon_k}(Q_i) \leftarrow \max_j \mathcal{E}_{\epsilon_k}(Q_i; Q_j)$;

    **end**

**end**

**Output:** Sort $\{Q_i\}_{i=1}^m$ in ascending order according to $\min_k \mathcal{E}_{\epsilon_k}(Q_i)$

---

**Remark on Footnote 4** As Footnote 4 mentioned, there is a corner case where our resolution selection rule fails: when $\epsilon_{\text{dct}} = \infty$, a constant $Q$ that predicts $\mathbb{E}_D[r]/(1-\gamma)$ can have 0 `BVFT-loss`. This can be prevented in theory by adding a penalty term $\propto \epsilon_{\text{dct}}$, though in practice we never observe such a degenerate behavior and hence did not include the penalty in our implementation.

## A.1 Details of `BVFT-PE-Q`

Here we provide the details about `BVFT-PE-Q`. As mentioned in Section 5, when $\pi_i \neq \pi_{Q_i}$, it is possible that $Q_i = Q^{\pi_i}$ but $\pi_i$ itself is a poor policy, in which case `BVFT-PE-loss`$((\pi_i, Q_i))$ is still 0. We address this issue by taking inspiration from a telescoping identity commonly used in the OPE literature [UHJ20]: $\forall(\pi, Q)$, let $d^\pi$ be the normalized discounted state-action occupancy, then

$$J(\pi) = \mathbb{E}_{s \sim d_0}[Q(s, \pi(s))] - \frac{1}{1-\gamma} \mathbb{E}_{d^\pi}[Q - \mathcal{T}^\pi Q].$$

This tells us that we can use *any* $Q$ to evaluate $\pi$, as long as we penalize its prediction $\mathbb{E}_{s \sim d_0}[Q(s, \pi(s))]$ with the Bellman error of $Q$ w.r.t. $\pi$. Inspired by this, we consider `BVFT-PE-Q`$((\pi_i, Q_i); \{(\pi_j, Q_j)\}_{j=1}^m) = $ `BVFT-PE-loss` $- \lambda \mathbb{E}_\mu[Q_i]$, where `BVFT-PE-loss` is a form of Bellman error w.r.t. $\pi$, and $\mathbb{E}_\mu[Q_i]$ (which is what AvgQ uses to rank the $Q_i$'s) is a variant of $\mathbb{E}_{s \sim d_0}[Q(s, \pi(s))]$. We are ranking policies in the descending order according to the loss, which is why the signs are the opposite to the telescoping identity. Intuitively, this avoids the degenerate issue because when $Q = Q^\pi$ for a poor $\pi$, `BVFT-PE-Q`$((\pi_i, Q_i)) = -\lambda \mathbb{E}_\mu[Q^\pi]$; instead of receiving a 0 loss as in `BVFT-PE`, the policy is still penalized for having a low $Q^\pi$. This new loss is effectively a linear combination of AvgQ and `BVFT-PE`, and $\lambda$ is an additional hyperparameter that determines their relative contributions.

# B Proofs

## B.1 Proof Sketch of Proposition 2

The basic idea is that $\mathcal{G}_{i,j}$ is piecewise-constant by design, and satisfies $Q^\star \in \mathcal{G}_{i,j}$ when $i^\star \in \{i, j\}$. Based on Proposition 1, this implies that whenever $i^\star \in \{i, j\}$, $Q = Q^\star \Leftrightarrow \|Q - \mathcal{T}_{\mathcal{G}_{i,j}} Q\|_{2,\mu} = 0$. Then, to show that $Q_i = Q^\star \Leftrightarrow$ `BVFT-loss`$(Q_i; \{Q_j\}_{j=1}^m) = 0$, it suffices to show that $Q_i = Q^\star \Rightarrow$ `BVFT-loss`$(Q_i) = 0$ and $Q_i \neq Q^\star \Rightarrow$ `BVFT-loss`$(Q_i) \neq 0$ :

- When $Q_i = Q^\star$, $Q^\star \in \mathcal{G}_{i,j}$ for any $j$, so `BVFT-loss`$(Q_i) = $ `BVFT-loss`$(Q^\star) = 0$.

- When $Q_i \neq Q^\star$,

$$\texttt{BVFT-loss}(Q_i) = \max_j \|Q_i - \mathcal{T}_{\mathcal{G}_{i,j}} Q_i\|_{2,\mu}$$

$$\geq \|Q_i - \mathcal{T}_{\mathcal{G}_{i,i^\star}} Q_i\|_{2,\mu} \neq 0. \qquad (Q^\star \in \mathcal{G}_{i,i^\star})$$

## B.2 Proof of Theorem 4

In this section we provide a full proof of Theorem 4. We start by quoting [XJ21]'s Assumption 1 on the data exploratoriness, which Theorem 4 relies on. We will also be able to reuse a number of their lemmas and propositions, due to the insensitivity of those results to the difference between $Q^\star$ and $Q^\pi$. That is, the proof holds literally by replacing any $\max_{a'} Q(s', a')$ term with $Q(s', \pi(s'))$ (or $\mathbb{E}_{a' \sim \pi(\cdot|s')}[Q(s', a')]$ if $\pi$ is stochastic), $Q^\star$ with $Q^\pi$, and $V^\star$ with $V^\pi$, in which case we say the proof "translates from $Q^\star$ to $Q^\pi$". Also note that their function class $\mathcal{F}$ corresponds to our $\{Q^l\}_{l=1}^L$.

**Assumption 1.** We assume that $\mu(s, a) > 0 \; \forall s, a$. We further assume that
(1) There exists constant $1 \leq C_{\mathcal{A}} < \infty$ such that for any $s \in \mathcal{S}, a \in \mathcal{A}, \mu(a|s) \geq 1/C_{\mathcal{A}}$.
(2) There exists constant $1 \leq C_{\mathcal{S}} < \infty$ such that for any $s \in \mathcal{S}, a \in \mathcal{A}, s' \in \mathcal{S}, P(s'|s,a)/\mu(s') \leq C_{\mathcal{S}}$. Also $d_0(s)/\mu(s) \leq C_{\mathcal{S}}$.
It will be convenient to define $C = C_{\mathcal{S}} C_{\mathcal{A}}$.

**Lemma 5** (Lemma 3 of [XJ21] translated to $Q^\pi$). *Let $\phi$ be a partitioning of $\mathcal{S} \times \mathcal{A}$, such that $\phi(s,a) = \phi(\tilde{s}, \tilde{a})$ means that $(s,a)$ and $(\tilde{s}, \tilde{a})$ are in the same partition. Define $M_\phi = (\mathcal{S}, \mathcal{A}, P_\phi, R_\phi, \gamma, d_0)$, where*

$$R_\phi(s,a) = \frac{\sum_{\tilde{s},\tilde{a}:\phi(\tilde{s},\tilde{a})=\phi(s,a)} \mu(\tilde{s}, \tilde{a}) R(\tilde{s}, \tilde{a})}{\sum_{\tilde{s},\tilde{a}:\phi(\tilde{s},\tilde{a})=\phi(s,a)} \mu(\tilde{s}, \tilde{a})}.$$

$$P_\phi(s'|s,a) = \frac{\sum_{\tilde{s},\tilde{a}:\phi(\tilde{s},\tilde{a})=\phi(s,a)} \mu(\tilde{s}, \tilde{a}) P(s'|\tilde{s}, \tilde{a})}{\sum_{\tilde{s},\tilde{a}:\phi(\tilde{s},\tilde{a})=\phi(s,a)} \mu(\tilde{s}, \tilde{a})}.$$

*Then $\mathcal{T}_{\mathcal{G}}^\pi$ for the piecewise-constant class $\mathcal{G}$ induced by $\phi$ is the Bellman operator for policy $\pi$ in MDP $M_\phi$.*

**Proposition 6** (Proposition 1 and Lemma 6 of [XJ21]). *Let $\nu$ be a distribution over $\mathcal{S} \times \mathcal{A}$ and $\pi$ be a policy. Let $\nu' = P(\nu) \times \pi$ denote the distribution specified by the generative process $(s', a') \sim \nu' \Leftrightarrow (s, a) \sim \nu, s' \sim P(\cdot|s, a), a' = \pi(s')$. Under Assumption 1, we have $\|\nu'/\mu\|_\infty := \max_{s,a} \nu'(s,a)/\mu(s,a) \leq C$. Also note that $\|(d_0 \times \pi)/\mu\|_\infty \leq C$. Furthermore, the same is true when the MDP dynamics is replaced by $M_\phi$ for any $\phi$.*

**Proposition 7** (Proposition 5 of [XJ21] translated to $Q^\pi$). *Given any $\phi$, define $\epsilon_\phi := \min_{g \in \mathcal{G}} \|g - Q^\pi\|_\infty$, where $\mathcal{G}$ is the piecewise-constant class induced by $\phi$. Fixing any $\epsilon_1, \tilde{\epsilon}$. Suppose the dataset $D$ has size*

$$|D| \geq \frac{32 V_{\max}^2 |\phi| \ln \frac{8 V_{\max}}{\tilde{\epsilon}\delta}}{\tilde{\epsilon}^2} + \frac{50 V_{\max}^2 |\phi| \ln \frac{80 V_{\max}}{\epsilon_1 \delta}}{\epsilon_1^2}, \qquad (4)$$

*where $|\phi|$ is the number of partitions in $\phi$ and $V_{\max} = R_{\max}/(1 - \gamma)$. Then, with probability at least $1 - \delta$, for any $\nu \in \Delta(\mathcal{S} \times \mathcal{A})$ such that $\|\nu/\mu\|_\infty \leq C$,*

$$\|f_0 - Q^\pi\|_{2,\nu} \leq \frac{2\epsilon_\phi + \sqrt{C}(\|f_0 - \widehat{\mathcal{T}}_{\mathcal{G}}^\pi f_0\|_{2,D} + \epsilon_1 + \tilde{\epsilon})}{1 - \gamma}, \qquad (5)$$

*where $\widehat{\mathcal{T}}_{\mathcal{G}}^\pi$ is the empirical estimation of $\mathcal{T}_{\mathcal{G}}$ based on dataset $D$. At the same time,*

$$\|f_0 - \widehat{\mathcal{T}}_{\mathcal{G}}^\pi f_0\|_{2,D} \leq (1 + \gamma)\|f_0 - Q^\star\|_\infty + 2\epsilon_\phi + \tilde{\epsilon} + \epsilon_1. \qquad (6)$$

*Proof.* This is a central result of [XJ21]'s proof and we show that the entire proof translates to $Q^\pi$. In particular, the result relies on a number of lemmas in [XJ21], among which

- Lemmas 6 (quoted above), 8, and 10 (two concentration bounds) are independent of $Q^\star$ or $Q^\pi$ and can be used without modification.

- Lemma 9 provides concentration bound for $\widehat{\mathcal{T}}_{\mathcal{G}} f$ for a fixed $f$, and only uses the boundedness of $V_f(s') := \max_{a'} f(s', a')$. The same bound holds for $\widehat{\mathcal{T}}_{\mathcal{G}}^\pi$ when we replace $V_f$ with $f(s', \pi(s'))$.

Finally, the proof of the proposition itself translates when we replace $\pi_{f,f'}$ in their proof with simply $\pi$. This is because error propagation in policy evaluation is much simpler than that in policy optimization [FSM10; CJ19], so we only need to be concerned with the distributions generated by $\pi$ instead of various other policies (see e.g., Appendix I of [UIJKSX21]). $\square$

The final lemma we need is:

**Lemma 8** (Lemma 11 of [XJ21] translated to $Q^\pi$). *Let $\phi$ be the partitioning of $\mathcal{S} \times \mathcal{A}$ that induces $\mathcal{G}_{i,j}$ in BVFT-PE, satisfies $|\phi| \leq (V_{\max}/\epsilon_{dct})^2$. Let $l^\star := \arg\min_l \|Q^l - Q^\pi\|_\infty$. When $l^\star \in \{i, j\}$, we further have $\epsilon_\phi \leq \epsilon_{dct} + \min_l \|Q^l - Q^\pi\|_\infty$.*

***Proof of Theorem 4.*** The majority of the proof of [XJ21]'s Theorem 2 translates, except for the final part where they use $\|\hat{Q} - Q^\star\|$ (our $\hat{Q}$ is their $\hat{f}$) to bound the suboptimality of the greedy policy of $\hat{Q}$. However, since we are only concerned about policy evaluation, we are not interested in this part that does not translate. What is useful to us is an intermediate result that translates: when the sample size is set to

$$|D| \geq \frac{82 V_{\max}^4 \ln \frac{160 V_{\max} L}{\tilde{\epsilon}\delta}}{\tilde{\epsilon}^2 \epsilon_{dct}^2}, \tag{7}$$

with probability at least $1 - \delta$, for any $\nu$ s.t. $\|\nu/\mu\|_\infty \leq C$,

$$\|\hat{Q} - Q^\pi\|_{2,\nu} \leq \frac{(2 + 4\sqrt{C})\epsilon_{\text{approx}} + 4\sqrt{C}(\epsilon_{dct} + \tilde{\epsilon})}{1 - \gamma}.$$

To guarantee that $\frac{4\sqrt{C}(\epsilon_{dct}+\tilde{\epsilon})}{1-\gamma} \leq \epsilon V_{\max}$, we set $\epsilon_{dct} = \tilde{\epsilon} = \frac{\epsilon R_{\max}}{8\sqrt{C}}$. Plugging this back into Eq.(7) yields the sample complexity in the theorem statement. $\square$

## C  Experiment Setup Details

### C.1  Environments

**Atari Games**  Atari games (or formally the Arcade Learning Environment [BNVB13]) are a set of arcade games used for benchmarking RL algorithms. The observations are raw-pixels and the action space is finite. We select 5 (Pong, Breakout, Asterix, Seaquest, and Space Invader) commonly used environments for our experiments based on their different characteristics. The pixel observation is resized to an $84 \times 84$ image per frame and with a frameskip of 4 and sticky action (25% of chance that the previous action will be executed instead of agent's action). Pong and Breakout have deterministic transition dynamics while the other 3 environments are more stochastic. In order to extract useful features from the high dimensional pixel input, we used multiple layers of convolutional neural networks followed by fully connected layers as our function approximator, which is the standard architecture in this domain [Mni+15].

**Classic Control and Box2D**  OpenAI gym [Bro+16] classic control provides many classical control problems. Among them, Cartpole and Acrobot have low dimensional (4, 6) continuous state space with discrete action space of cardinalities 2 and 3. Pendulum originally has a continuous action space but was modified to have 2 actions of swinging to left and right to match the other environments. All the classic control environments have deterministic transition dynamics.

We also experimented with LunarLander, a Box2D environment with higher complexity and difficulty than the classic control environments. The state space is continuous with discrete actions of firing engines at different locations to control the descent of the lunar lander. The episode stops when the lunarlander's pads touch the surface and rewards are given based on fuel consumption and landing location. We used simple 3-layer MLP to be the function approximator which is sufficient for the relatively low input dimension. The environment has random starting locations and velocities while the transition dynamics are deterministic.

**Mujoco**  The gym MuJoCo has a collection of continuous control tasks implemented based on the MuJoCo Simulator [TET12] and has been a popular testing environment for continuous-action RL algorithms. We included HalfCheetah, Hopper, and Walker2D to evaluate the performance of BVFT and other baselines.

| Hyperparameter | Atari | MuJoCo |
|---|---|---|
| Training size | 1M | 500K |
| Hidden size | FC: 256, 1024; CNN: (32,64,3136) | 64, 1024 |
| # hidden layers | FC: 1, 2 | 2, 3 |
| Learning rate | 0.0000625, 0.00001 | 0.001, 0.00001 |
| Learning steps | {200k:100k:1M}/{50k:50k:400k}/{50k:50k:400k} | {50k:25k:300k} |
| Algorithms | DQN/BCQ/CQL | BCQ/CQL |

| Hyperparameter | Taxi | Gym Control | Box2d |
|---|---|---|---|
| Training size | Online | Online | Online |
| Hidden size | Tabular | 64, 256, 1024 | 64, 256, 1024 |
| # hidden layers | Tabular | 2, 3 | 2, 3 |
| Learning rate | {5e-3:5e-3:2.5e-2} | 2.5e-4, 5e-4 | 2.5e-4, 5e-4 |
| Learning steps | {200k:50k:500k} | {50k:25k:250k} | {100k:50k:400k} |
| Algorithms | Q-learning | DQN | DQN |

Table 1: Hyperparameters of the candidate models. We consider an array of hyperparameters on the optimizer and the neural architecture to produce a diverse set of candidate models for selection. The notation $\{a : x : b\}$ is a shorthand for $\{a, a + x, a + 2x, \ldots, b\}$.

**Taxi** Taxi [Die00] is a classical reinforcement learning environment with a 2D grid world that simulates taxi driving along with the grids. At each time step, the taxi can choose among 6 actions of moving north, south, west, east, or stay and picks up or drops off the passenger. The RL agent will receive a reward of 20 if the taxi successfully picks up or drops of a passenger, otherwise the agent receives a reward of -1 at each time step. The original taxi environment has a grid size of $5 \times 5$, resulting in 500 total states ($25 \times 4 \times 5$, corresponds to 25 possible taxi locations, 4 destination locations, and 5 passenger locations(4 spawn locations and 1 in the taxi)). In order to further investigate the effect of randomness in transition dynamics, we performed additional experiments on a modified version of the original taxi environments with additional stochasticity: we replaced the only original passenger with 4 passengers randomly spawning and disappearing at the 4 locations every time step. The resulting environment has a total of 2000 states ($25 \times 2^4 \times 5$, corresponds to 25 possible taxi locations, $2^4$ passenger appearance status, and 5 taxi status). In addition, we added a parameter $p_{rand}$ to the environment such that at each time step, the taxi has a probability of $p_{rand}$ that it will act randomly instead of following the agent's action.

A main reason for considering this simple environment is the easy access to $\mathcal{T}$ and $Q^\star$ due to the tabular nature of the environment. This allows us to introduce a number of skylines to evaluate BVFT's performance, which would not be possible in more complex environments. We use tabular Q-learning as training algorithms to generate the candidate $Q$'s. The results of this study can be found in Figure 11.

## C.2 Training algorithms and datasets

BCQ [FCGP19] and CQL [KZTL20] are used to learn $Q$-functions in Atari from the RLUnplugged datasets [Gul+21]. (In Appendix D we also use online DQN to generate candidate policies in Atari.) For the classic control and Box2D domains, we deployed DQN [Mni+15] with 2/3 layer MLP as function approximators to learn the candidate model in an online manner. The offline dataset for policy selection is generated by first training an expert policy, and then mixing 70% of expert trajectories with 30% $\epsilon$-greedy trajectories w.r.t. the expert for $\epsilon = 0.5$. In MoJoCo, policies are learned using the continuous-action version of BCQ [FMP19] using D4RL datasets [FKNTL21]. In taxi, standard Q-learning is used to learn $Q^*$ online. See Table 1 for further details about the candidate hyperparameters of each training algorithm in each domain.

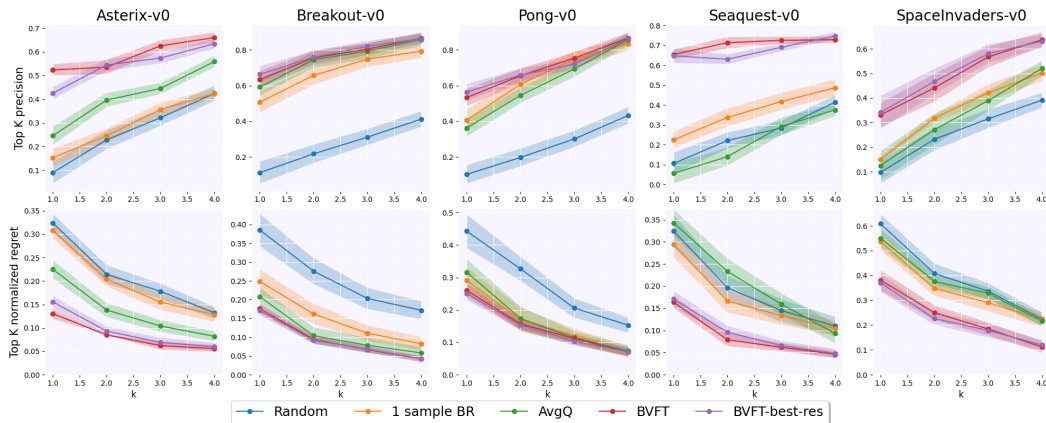

Figure 7: Policy selection in Atari with candidate policies learned by CQL. Results are qualitatively similar to when training algorithms are BCQ (Figure 2).

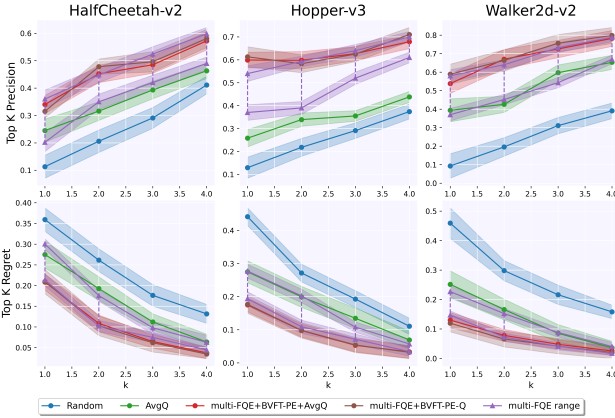

Figure 8: Policy selection in Mujoco using `BVFT-PE` and `BVFT-PE-Q` with candidate policies learned by CQL. Results are qualitatively similar to Figure 6L).

# D Additional Experiments

**Atari with CQL and DQN as the training algorithms** Figure 7 shows the results in Atari when we use CQL as the training algorithm (still using offline data), and Figure 9 shows what happens when DQN learns candidate policies using online interactions with the environment. Both results are qualitatively similar to Figure 2, where BCQ is used as the training algorithm. This demonstrates the robustness of BVFT w.r.t. different training algorithms, or even offline vs. online training.

**Mujoco with CQL as the training algorithm** Similarly, we also reproduce our results in Figure 6L with CQL as the training algorithm in Mujoco, and the results are qualitatively similar. See Figure 8.

**Robustness w.r.t. data exploratoriness** We test the sensitivity of different hyperparameter-free methods to data exploratoriness in two Atari domains. For this specific experiment, we generate our own data (unlike the other Atari experiments which use RLUnplugged datasets) by training an expert policy and mixing it with different probabilities of taking actions randomly. Figure 10 shows the robustness of BVFT across different amount of randomness in the data-generating policy. Similar robustness can be observed in Taxi (Figure 11, middle vs. right).

**Sample efficiency** Figure 12 plots the performance of different methods against sample size for policy selection in Atari. Note that BVFT is competitive in the small-sample regime, and is the only method that improves when more data becomes available.

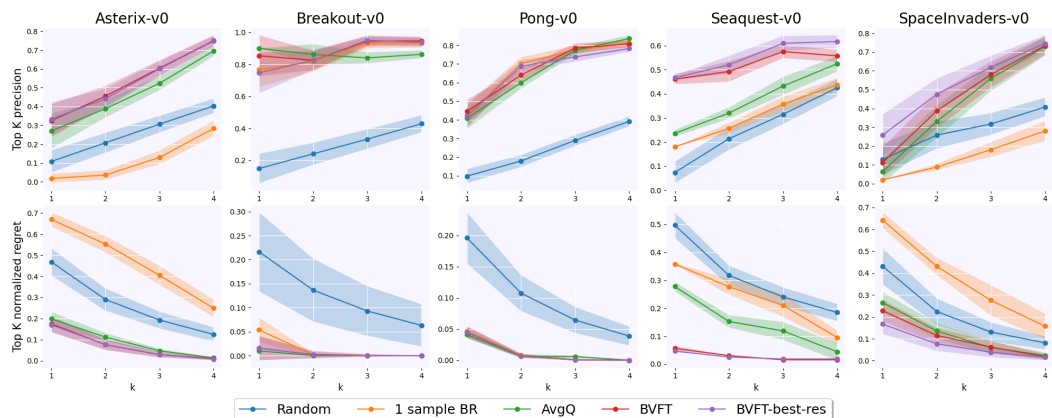

Figure 9: Policy selection in Atari with candidate policies learned by (online) DQN.

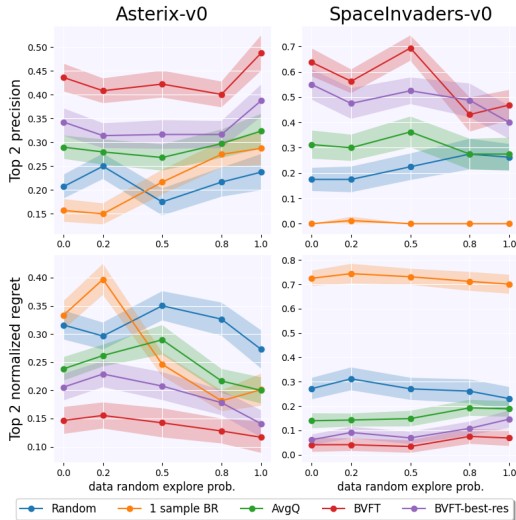

Figure 10: Sensitivity of policy-selection methods against data exploratoriness in 2 Atari games. X-axis is the probability of choosing a random action instead of the expert policy when generating the offline dataset for policy selection. Notice BVFT's performance is not sensitive to the level of stochasticity in the data-generation policy.

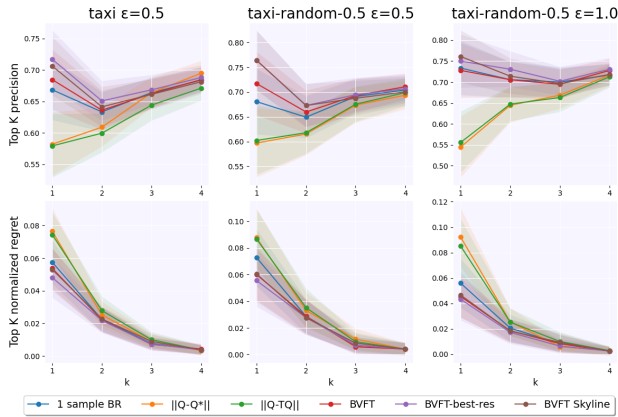

Figure 11: Policy selection in Taxi (left) and its modified version with additional stochasticity in transition dynamics (middle and right). $\epsilon$ indicates the stocahsticity in the data-generating policy. "BVFT Skyline" is similar to BVFT, except that $\mathcal{G}_{i,j}$ is produced by discretizing $\mathcal{S} \times \mathcal{A}$ according to $Q^\star$ itself instead of the compared functions. $\|Q - Q^\star\|$ and $\|Q - \mathcal{T}Q\|$ are skylines that use the knowledge of $Q^\star$ and $\mathcal{T}$, which are usually not available. BVFT competes favorably and sometimes outperforms these skylines. Baselines "Random" and "AvgQ" are not included as their performance is much worse and off the chart. Middle and right figures also show the robustness of the result w.r.t. the data-generation policy.

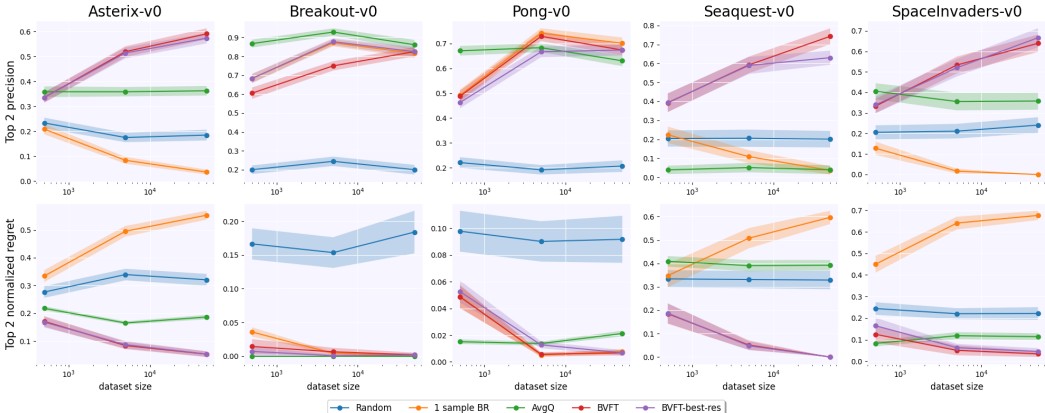

Figure 12: Top-2 precision and regret vs data size in 5 Atari environments. BVFT's performance improves as the data size increases while other baselines stays relatively constant.