# OpenReview forum: "Towards Hyperparameter-free Policy Selection for Offline Reinforcement Learning"
_NeurIPS.cc/2021/Conference — NeurIPS 2021 Poster_

### Official Review · Reviewer_8caC · 2021-07-16

**Rating:** 9
**Confidence:** 3

**Summary:**

Offline Policy Evaluation (OPE) is when one needs to choose the best policy within a set of policies given only access to an offline dataset. Most OPE algorithms relies on function approximation and thus has hyperparameters. Previous works showed promising performance when the hyperparameters are tuned “online”, thus limiting their use case in practice. This paper proposes a way to select hyperparameters using purely the given offline dataset.

This paper expands on BVFT and turns it into a practical OPE algorithm without hyperparameters. More specifically, it proved that the BVFT-loss in its limit is equivalent to the naive estimator of the Bellman residual (which is biased in a stochastic environment). When one chooses a reasonable discretization resolution, experimentally BVFT performs better than the naive estimator. When the environment is continuous, the authors proposed to combine BVFT with FQE or other hyperparameter-dependent OPE methods. The experiments showed that the proposed method often matches or outperforms existing OPE algorithms.


**Limitations And Societal Impact:**

The authors discussed the limitations of their method and either remedies or future research directions in the paper.

**Main Review:**


The paper is written with exceptional clarity, where the authors provided both theoretical and experimental results backing their claim. Since there is no prior work on hyperparameter-less OPE, the authors chose a reasonable way to compare with other OPE baselines, where the baseline hyperparameters are finetuned on one set of environments and evaluated on another. The proposed method also took into consideration practical cases, such as 1. the state or action space is continuous 2.$(\pi, Q)$ pair is not trained till the optimum, where either the policy is not greedy w.r.t. Q, or the Q is under-trained before the policy converges.

This is a significant work that will allow OPE to have more wide-spread use in practice where hyperparameter sweeps are expensive or infeasible. It is novel both compared to BVFT and to other OPE algorithms, providing both theoretical guarantees and practical flexibility (e.g. giving the option of BVFT-PE and BVFT-PE-Q depending on whether the user is able to tune one extra hyperparameter either online or on a similar environment.)

Minor nit-picking:
1. l88. ... to test more than 1 policies -> … to test more than 1 policy



**Time Spent Reviewing:**

8

---

> ### Author Response · Authors · 2021-08-05
> **Initial response to review**
>
> Thanks for your appreciation of our paper! We are also extremely excited about the possibility that these methods---after further developed---could become part of the standard pipelines in the application of offline RL to real-world problems!
>
> - The experiments showed that the proposed method often matches or outperforms existing OPE algorithms.
>
> Just out of caution, we would like to emphasize that our goal in Section 5 is _not_ to outperform existing OPE algorithms, since we rely on them to generate high-quality Q-functions (when their hyperparameters are set right), and being able to match the _best_ OPE instance is already a strong positive result. The fact that we actually slightly outperforms the best OPE instance in experiments is actually a bit surprising to us, which implies different policies may be better evaluated by different OPE algorithms and our methods can identify the best match in an adaptive manner.

---

### Official Review · Reviewer_bSdg · 2021-07-16

**Rating:** 6
**Confidence:** 1

**Summary:**

The paper proposed a new method Batch Value-Function Tournament (BVFT) for the off-policy evaluation of RL. The algorithm is hyperparameter-free thus could be applied to a variety of Offline RL algorithms. Experiments are conducted on both discrete and continuous control benchmarks. BVFT outperforms other baselines on some Atari (discrete control) environments but cannot match OPE methods on the Mujoco (continuous control) environments. The authors further consider a method combining BVFT and OPE. This method is able to tie with the performance OPE on Mujoco and has some nice properties. In addition to empirical results, the authors also provide theories to support their method.


**Ethical Concerns:**

No ethical concerns from my side.

**Limitations And Societal Impact:**

In section 6 the authors have addressed most of the limitations of their work. One additional limitation is that their method would mostly be restricted to policy selection for Q-learning based algorithms. Though this includes a large family of offline RL approaches, imitation learning based, or model based algorithms also achieve promising results in the offline RL problem, and are worth studying. The authors could try to discuss how to extend BVFT on addressing these algorithms.

The BVFT algorithm differs from previous OPE methods in the idea and implementation, and achieves SOTA performance on discrete control benchmarks. But to avoid performance degradation in continuous control, the authors combine BVFT and OPE. Despite some advantages over OPE, this introduces additional complexity into the implementation and makes it unclear which part contributes to a promising policy selection. As for offline RL algorithms, the need for distributional constraint or value pessimism is concentrated on continuous control tasks. The situation is more interesting here, and the paper could have provided more insight. Nevertheless, it would be more significant if there is a unified approach that performs consistently well across discrete and continuous control benchmarks.

I don’t see any potential negative societal impact of this work.



**Main Review:**

Off-policy evaluation of RL is indeed an important open question for Offline RL. The authors address it with a novel method, with extensive comparison to the existing OPE methods and some other natural approaches based on quantities like Bellman Residual, Average Q value, etc.

The clarity of this paper is ok but could be improved. I appreciate the thoughtful writing with little typos. The authors often try to ask questions then answer them. But sometimes the answer is not supported with convincing evidence.

Moreover, the authors provide some theoretical properties of their BVFT algorithm. In particular, Proposition 2 which proves the equivalence between BVFT-loss and Bellman residual is interesting. However, it is not very clear how this relates to the BVFT in implementation (I see the authors provide some discussion, but giving more augmentation or insight would enhance the flow of logic).

Here are some confusions/suggestions:
On the experiment part, using the 2 quantities top-k regret and top-k precision as the evaluation score is straightforward and considerate. But it would be more clear if the authors provide a discussion on their (different) meaning alongside the definition. (I would have some explanations on these quantities but I am not sure if the authors have the same thought.)
To study the variance, the authors also consider the randomness introduced by datasets. It would also be good to consider randomness introduced by the algorithm training process (initialization and SGD).
BCQ and CQL are both experimented on the Atari benchmark, but I wonder why the authors just consider BCQ on Mujoco? These 2 offline RL algorithms are substantially different and it would be interesting to see comprehensive results of both.
Some of the algorithmic discussion in the appendix could be moved to the main paper, while some derivations in Sections 3 and 4 could be moved to the appendix, if they are not essential for understanding the main idea of the algorithm. It would also be good if the authors provide more discussion on the intuition of their approach.

*post-rebuttal*:
I thank the authors for their detailed reply. I think it would be good for to include

I think this paper studies an important question and is illuminating in some sense. I consider raising the score if the above are addressed.

*


**Time Spent Reviewing:**

12

---

> ### Author Response · Authors · 2021-08-05
> **Initial response to review**
>
> Thanks for your valuable feedback and comments! We address the major concerns below:
>
> - __their method would mostly be restricted to policy selection for Q-learning based algorithms.__
>
> Not exactly. Our methods in Section 5 are general and can be readily applied to any training algorithms. While our initial formulation assumes access to (π,Q) pairs, we discard Q in Step 2 (L269) and re-fit it using OPE. Therefore, in our final approach, the training algorithms do not need to provide Q to begin with and only need to produce the candidate policies! That said, we admit that this is a subtle point and may not be easily recognized from our writing. We will clarify this in revision.
>
> - __[Combining BVFT with OPE] tie with the performance OPE on Mujoco… Despite some advantages over OPE, this introduces additional complexity into the implementation and makes it unclear which part contributes to a promising policy selection.__
>
> We respectfully disagree. First, most OPE methods come with hyperparameters themselves (L283), which is a problem that no practitioner can avoid and cries for a solution. When reviewer says our methods “tie with OPE”, they are considering the _best_ OPE instance, but this comparison is unfair because such a performance is obtained by “cheating”--- we know how different OPE instances works because the simulator gives us the groundtruths, which is not available in real-world applications that need offline RL. In fact, as Figure 5R shows, the best OPE hyperparameters for the 3 Mujoco domains are not even the same, with orange performing the best in Hopper but considerably worse than blue in Walker2d. Our method can automatically select the best hyperparameters for OPE in different domains adaptively, therefore complements existing OPE methods and addresses an important open problem in them.
>
> Second, regarding “additional complication”, we want to emphasize that after running OPE with different hyperparameters, our own methods (BVFT-PE or BVFT-PE-Q) are very simple and all the computations are in closed-form, so the additional complexities are very limited.
>
> - __Top k regret and precision__
>
> These are standard metrics used in recent policy selection works, but we agree some discussions of their meanings will be helpful. Indeed, we have alluded to the meaning of top-k regret in L87-89: after policy selection, we may be able to test k policies _online_ and identify the best policy among them, and top-k regret measures the regret compared to the best policy which may fail to be captured in our top-k list. We will expand this discussion. To this end, we think the top-k regret is more meaningful and directly related to real-world application scenarios, and top-k precision is secondary and plays a more assistive role in our evaluation. (After all, they are mostly consistent with each other.)
>
> - __Proposition 2 which proves the equivalence between BVFT-loss and Bellman residual… how this relates to BVFT in implementation?__
>
> Proposition 2 gives the basic definition and property of BVFT-loss; we guess you actually mean Proposition 3? As mentioned in Section 4, this proposition provides important hints of how to select the discretization resolution (which we were _clueless_ about when the project started): BVFT-loss is equivalent to 1-sample BR when discretization approaches 0, and we know that 1-sample BR is _positively biased_. This provides theoretical justification for minimizing BVFT-loss over the discretization resolutions, which is also supported by empirical evidence (Figure 1R).
>
> - __To study the variance, [authors should also] consider randomness [in] the training process (initialization and SGD).__
>
> This is a good point, and our results have partially accounted for this. As Line 119 mentions, in each trial, not only we randomly sample the dataset for policy selection, but also sample a subset of the candidate policies/value-functions from a larger pool (the pool usually has 50-60 models, corresponding to the combinations in Table 1). Each candidate in the pool is trained with random initialization and fresh SGD noise, so such variance does affect the final result.
>
> - __Only BCQ on Mujoco?__
>
> We completely agree it is important to study how sensitive our method is to different training algorithms. Given limited computational resources, we decided to demonstrate this in Atari games. Indeed, besides CQL and BCQ, we even considered _online_ DQN on Atari (Figure 7 in Appendix); in this case the training algorithm is not even offline (see L106 for why this is still practically relevant), yet the results are still qualitatively the same. We believe this has sufficiently demonstrated the robustness of our algorithm. That said, according to your suggestion we will run additional experiments using CQL on Mujoco, but this will take quite some time given the limited computational resources we have access to and other real-life constraints. Our current plan is to reproduce Figure 5L for CQL, but if you have other suggestions please let us know!

---

> > ### Comment · Reviewer_bSdg · 2021-09-07
> > **Response to rebuttal**
> >
> > Thanks for your detailed feedback. Since your response clarified some of my concerns, I would raise my score to 6 presuming you would add the discussions in your response to the camera ready version.

---

### Official Review · Reviewer_qEm2 · 2021-07-22

**Rating:** 8
**Confidence:** 4

**Summary:**

Batch Value Function Tournament (BVFT) is an existing approach to offline policy selection that uses piecewise linear value function approximations and selects the policy with the smallest projected Bellmane error in that space. This paper automatically selects the BVFT value function resolution by using a clever argument involving the convergence of the projected Bellman error to the regular Bellman error when the resolution goes to zero. On datasets from Atari and Gym datasets this approach is very close to selecting the best resolution in retrospect. They also show how this approach can be extended to policy gradient methods for continuous actions, where the application of BVFT is less straightforward than in discrete action spaces.


**Limitations And Societal Impact:**

I appreciate the authors pointing out one of the major limitations of their work - “the success of BVFT relies on the existence of a reasonable approximation of Q* among the candidate functions”, and the fact that they explore (and to some extent address) the limitations in the context of continuous action spaces. And while I understand why this is not included in the paper, I definitely encourage the authors to try and produce results using the benchmarks in [Fu+21].

As mentioned in the main review, I think using this type of approach with clinical trial data for adaptive treatment regimes could have considerable positive societal impact.


**Main Review:**

To me the paper ranks quite highly in terms of originality - it can be seen as a relatively simple extension of an existing approach, but as I mentioned in the summary the extension is quite clever and seems to work really well on some problems and is well motivated. I also found the  paper very good in terms of quality and clarity

In terms of significance, I think the authors should explore more what kind of domains their approach can be really helpful in. My feeling is that it will be less useful in domains where complex function approximators are required for learning good policies, and their approach to continuous action space policy selection seems a bit complex for widespread practical use to me, plus it introduces lambda which is a hyperparameter that probably needs tuning (although the approach is interesting). I do think however that there is at least one important application area that looks perfectly suited for their approach, and that is OPE / policy selection from clinical trial data - see e.g. Raghu, A., Komorowski, M., Celi, L. A., Szolovits, P., and Ghassemi, M. (2017) Continuous state-space models for optimal sepsis treatment: a deep reinforcement learning approach. This is because the data there is limited but piecewise continuous value functions are likely to work well there, and also be more interpretable than neural approximations. I encourage the authors to explore this direction if they find it interesting.


**Time Spent Reviewing:**

4

---

> ### Author Response · Authors · 2021-08-05
> **Initial response to review**
>
> Thanks for your comments! Below are our responses to your comments:
>
> - __My feeling is that it will be less useful in domains where complex function approximators are required for learning good policies__
>
>
> - __[Proposed methods are particularly suited to medical applications] because the data there is limited but piecewise continuous value functions are likely to work well there, and also be more interpretable than neural approximations.__
>
> We feel there might be some misunderstanding here. The piecewise constant function classes are created _inside_ BVFT as part of the algorithm, and they are not given by users as inputs. The candidate functions themselves, on the other hand, __can be arbitrary functions__, and in our experiments, they are mostly generated from neural-network training using deep RL algorithms.
>
> More importantly, a major highlight of BVFT’s theory is that its sample complexity does _not_ depend on the complexity of the underlying MDP (say, size of state-action space, complexity of dynamics, etc) in any direct manner. Roughly speaking, this is because BVFT always does aggressive state-action aggregation that compresses the problem down to merely O(1/epsilon^2) state-action pairs (see Figure 1L and text on L165), regardless of the MDP complexity. This is precisely why BVFT is much more sample-efficient than FQE in Atari games, because FQE needs to use neural nets to learn good representation over the raw pixel observations, where BVFT---both in theory and in practice---is not affected by such complexity.
>
> To summarize, we guess your intuition is that it is relatively easy for users to come up with piecewise function classes to capture value functions in medical applications, and __you believe BVFT requires this condition to do well. That is not true.__ BVFT does not require the user to supply such a piecewise-constant function class, but rather creates them automatically and dynamically inside the algorithm, based on the candidate functions which themselves can be arbitrarily complex. Therefore, BVFT applies equally well to complex problems (such as Atari), and is not restricted to the kind of problems you described (if we understand you correctly).
>
> We understand that this is quite tricky because BVFT is a new (and somewhat sophisticated) theory that the community is not familiar with, and your comments are very helpful in letting us know what are common misconceptions; we will try to better clarify them in the revision.
>
> With all that said, we totally agree that medical application would be a great domains to apply offline RL to, and it would be definitely interesting to evaluate our methods in such domains in subsequent works.
>
> - __plus it introduces lambda which is a hyperparameter that probably needs tuning__
>
> Agree that this lambda parameter is not ideal in BVFT-PE-Q, though we would like to emphasize that the alternative method (Strategy 2 on L291) using BVFT-PE does not come with this lambda and is only slightly worse than BVFT-PE-Q.

---

> > ### Comment · Reviewer_qEm2 · 2021-08-05
> > **Re: Initial response**
> >
> > Hmm I am a bit surprised by the response, I don't think the review indicated that I believe that the piecewise constant functions need to be "given by users as inputs". I was just pointing out that it would be good for the authors to think more about the types of problems where this type of representation is likely to work well and the types where it's maybe not the best approach. Again, the paper itself says that “the success of BVFT relies on the existence of a reasonable approximation of Q* among the candidate functions”. Otherwise I don't disagree with the response, but it also doesn't alter my understanding of the paper.

---

> > > ### Author Response · Authors · 2021-08-05
> > > **Thanks for the quick feedback**
> > >
> > > Thanks for the feedback (that was fast :). We totally agree that we need to "think more about the types of problems where this type of representation is likely to work well and the types where it's maybe not the best approach".
> > >
> > > That said, we still feel there might be some subtle technical misunderstandings. Let us elaborate a bit more and see if we misunderstood your point:
> > >
> > > 1. "the success of BVFT relies on the existence of a reasonable approximation of Q* among the candidate functions"
> > >
> > > In complex environments, this can be guaranteed by training highly expressive neural networks on a large amount of data. There are application domains that fit this scenario. For example, in online recommendation systems where decisions have delayed consequences, the company usually has large amounts of data but testing on real user traffic is difficult and requires many levels of internal approval, where offline RL is a good fit.
> > >
> > > 2. Perhaps the following proposition can help: for every MDP, there exists a piecewise constant function class (based on a partitioning of SxA up to O(1/\epsilon) partitions) that represents Q* accurately up to \epsilon error. The proof is trivial because you can just aggregate the state-action pairs according to the true Q* value. This proposition does not require any additional assumptions: size of S and A, MDP dynamics, rewards, all of these do not matter, and this observation underlies the derivation of the original BVFT paper. Note however, that if one has a pre-defined topology/metric over the state-action space, such a function class will not necessarily respect the topology (i.e., it may aggregate states and actions all over the state-action space, regardless of whether they are "close" or "far" under the topology).
> > >
> > > So to summarize, the MDP does not need to bear additional structure for such classes to exist, and therefore related structures (e.g., topology/metric of S and A under which Q* is smooth or has other nice properties) do not affect the applicability of BVFT.
> > >
> > > If this is not what you have in mind, sorry we misunderstood you, but hopefully this clarification is helpful!

---

> > > > ### Comment · Reviewer_qEm2 · 2021-08-18
> > > > **Re:**
> > > >
> > > > Apologies for responding much more slowly the second time around. What I ultimately want to understand is how the piecewise continuity assumption, weak as it may be, affects the types of problems where we might expect this approach to outperform vs underperform using FQE with a well-tuned net.

---

> > > > > ### Author Response · Authors · 2021-08-18
> > > > > **Thanks for your comment**
> > > > >
> > > > > Thanks for your comments and efforts on trying to understand the paper and its limitations! We really appreciate it.
> > > > >
> > > > > - What I ultimately want to understand is **how the piecewise continuity assumption, weak as it may be,** affects the types of problems
> > > > >
> > > > > That's exactly the confusion we were trying to clarify through the previous responses. Roughly speaking, the piecewise constant (not continuity) property is _not_ even an assumption. Note that we, as well as the original work of Xie & Jiang, never state this as an assumption of the underlying MDP; it is merely a concept used in the **internal mechanism** of the algorithm. Therefore, it has no bearing on the type of problems our method can apply to (since it is not an assumption to begin with).
> > > > >
> > > > > Maybe it could help to give a somewhat extreme example to further clarify this issue. Suppose the state space is $\mathbb{R}$, and the function of interest $f(x)$ is a highly weird/non-smooth one: it takes value $0$ when $x$ is a rational number, and $1$ when $x$ is irrational. Crazy as this function seems, in the context of our paper, we still say that it is "piecewise-constant", where all the rational numbers belong to 1 piece, and all the irrational numbers belong to the other piece. We believe that the confusion may partially come from how we (and the original theory work) use the term "piecewise-constant" in a way different from common conception. In the above example, for instance, most people wouldn't call the function "piecewise-constant" because they would assume that a "piece" has to be a continuous interval on the real number line. As we mentioned in previous responses, however, our approach does not assume such a topology and metric of the state space, and by not relying on it, our use of "piecewise-constant" function classes is not really an assumption.
> > > > >
> > > > > Following this logic, you can see that as long as the function takes finitely many possible values, we can always say that it belongs to a piecewise constant function class where the number of pieces = number of possible values the function can take, even if the state-action space is high-dimensional. When the number of possible values of the function is infinite, a discretization step would be needed, and we discuss how to select the discretization resolution in the paper. This is precisely why our method scales so well and is sample-efficient in Atari games which has high-dimensional pixel observations.

---

> > > > > > ### Comment · Reviewer_qEm2 · 2021-08-19
> > > > > > **Re:**
> > > > > >
> > > > > > Thank you for the in-depth attempts to clarify this, I appreciate them. But it seems to me like whether (1) and (2) in Proposition 1 are satisfied is not guaranteed, and of course this method is not going to provide a perfect oracle in general. Can the authors point to any type of problem where it might struggle? Also, I want to emphasize that I do like the approach and I do think this is a very good paper overall and would definitely want to see it accepted, I was just wondering if the authors have any additional thoughts on applicability beyond what is mentioned on lines 335-341, which is already quite good.

---

> > > > > > > ### Author Response · Authors · 2021-08-19
> > > > > > > **Proposition 1 and Applicability**
> > > > > > >
> > > > > > > Thanks for the questions.
> > > > > > >
> > > > > > > Edit: typo in the first sentence ("do always hold" => "do not always hold")
> > > > > > >
> > > > > > > (1) and (2) in Proposition 1 do not always hold: This is exactly what we said on Line 139 ("The problem is that it is very difficult to find G that satisfies the above 2 criteria"), which would indeed be the case "if we leave the design of G to the user" (Line 140). The "magical" part of the theoretical result of [XJ20] is that they show how one can design G satisfying (1) and (2) in Proposition 1 automatically using nothing but the candidate Q-functions themselves.
> > > > > > >
> > > > > > > In the simple case of 2 candidate functions (one of which is Q*), we illustrate how to do this on Lines 143-145 and Figure 1L. The resulting $\mathcal{G}$ satisfies (1) and (2), which one can verify relatively easily: (1) it is piecewise constant by construction, and (2) $Q^\star \in \\{Q_1, Q_2\\} \subset \mathcal{G} \Rightarrow Q^\star \in \mathcal{G}$, which is also by construction. The case of multiple functions (m>2) is more complicated and involves a tournament procedure that runs the 2-function process on every pair of functions, and the $\mathcal{G}$ dynamically changes with the pair of functions being compared (see $\mathcal{G}_{i,j}$ in Eq.2). The underlying analysis is more involved so we omit in the discussion since this paper is targeted at a broader audience. Suffice to say that the conditions and guarantees for m>2 is similar: the algorithm succeeds as long as one of the candidate functions is (approximately) $Q^\star$. You can also check the original [XJ20] paper: they only make two assumptions, one of which is about exploratory data, and the only remaining assumption is Q* approximately lying in the set of candidate functions (this is called the realizability assumption in the context of training, hence their title "with only realizability").
> > > > > > >
> > > > > > > Hopefully at this point you are seeing why "piecewise-constant" is an internal mechanism of the algorithm and does not impose any kind of constraints on the underlying MDP, and the real challenge is to handle the case where none of the candidate functions is $Q^\star$, which we spend the whole Section 5 on. Another challenge is the lack of exploratory data, which we briefly touch on Line 342, though in the experiments we did not find the data coverage to be a particular problem. To summarize, if training method can successfully fit a good value function under the right set of hyperparameters and the data provides sufficient coverage (which are not always guaranteed), then our method will succeed, which is demonstrated theoretically and empirically by the paper.

---

> > > > > > > > ### Author Response · Authors · 2021-08-19
> > > > > > > > **Follow-up**
> > > > > > > >
> > > > > > > > Please be minded that we corrected a typo in the previous comment. Also to add: among all the $\mathcal{G}_{i,j}$ created in the algorithm, many of them will not satisfy (2) in Proposition 1 because $Q^\star$ is not among the pair of functions compared (since most likely $Q^\star$ will only appear in the candidate set once if at all). However, this is fully accounted for by the tournament procedure, and to quote [XJ20] (pg 4): "Careful readers may wonder what happens when $Q^\star \notin \\{Q_i, Q_j\\}$, as [ $Q^\star$ being one of the compared functions ] is obviously violated. As we will show ..., the outcomes of these “bad” comparisons simply do not matter..."
> > > > > > > >
> > > > > > > > We did not delve into these details because explaining this in full clarity to a non-theory audience (which we target at) is very challenging. All we can do is to provide a short summary to give the readers a taste of the theory involved. To really understand the theory behind the algorithm, perhaps reading the original work of [XJ20] is necessary.

---

> > > > > > > > > ### Comment · Reviewer_qEm2 · 2021-09-07
> > > > > > > > > **Re:**
> > > > > > > > >
> > > > > > > > > Thank you for all the discussion and clarification. I am assuming some of this can be included in the camera ready copy, and have increased my score to 8.

---

### Official Review · Reviewer_s3ZC · 2021-08-02

**Rating:** 7
**Confidence:** 1

**Summary:**

This paper focuses on a hyperparameter-free policy selection for offline reinforcement learning. The authors suggests two methods: (1) a automatic value-function selection method for discrete action spaces using Batch Value Function Tournament (BVFT), (2) a combination of OPE algorithms and BVFT variants for continuous action spaces (BVFT-PE, BVFT-PE-Q, ...)

**Limitations And Societal Impact:**

Yes, the authors adequately addressed the potential negative societal impact of their work.

**Main Review:**

Offline policy selection is an important problem for deploying offline reinforcement learning algorithms to many practical real-world problems. However, previous methods using off-policy policy evaluation (e.g. FQE) also require hyperparameter tuning, which makes a chicken-and-egg situation. This paper tackles this issue in a principled way using BVFT, which is the value function selection method with theoretically advanced Bellman error estimates. This paper has strong motivations and the methods are appropriate. Hence, I vote for the acceptance to this paper. I have following questions for this paper:

1. In line 267, BVFT-PE-Q loss uses a hyperparameter $\lambda$ which adjusts a penalization coefficient of the average critic value. (maybe hence the authors call their methods as "nearly"-hyperparameter-free in Section 6) How is $\lambda$ choiced for the experiment and how sensitive?

2. How many FQE instances are used for multi-FQE experiments (Figure 5L)? (maybe described as $L$ in the paper). I think it will be good to show that a choice of L does not significantly affect the result of experiment.




**Time Spent Reviewing:**

6

---

> ### Author Response · Authors · 2021-08-05
> **Response to review**
>
> Thanks for your comments! Here are our responses to your questions:
>
> 1. How is λ tuned and how sensitive: Line 300 mentions that λ is tuned on Hopper and the value is used without tuning on the other 2 Mujoco domains. According to our experience the method is not very sensitive to λ, though we did not carry out more thorough experiments on this matter. We would also like to emphasize that the other method in Section 5 based on BVFT-PE does not require such a λ parameter and performs only slightly worse.
>
> 2. We did L=4 and the 4 instances correspond to the legend of Figure 5R (this is mentioned on Line 294), where the instances differ in the number of hidden nodes in the neural architecture and the learning rate (lr). We did not do larger L because this experiment is somewhat expensive as we have to use 4 different OPE instances to evaluate a large pool of policies using neural nets. If you think it is important to observe the effect of different L, we can add this to our experiment todo-list and try to run it before the camera-ready if the paper is accepted.

---

> > ### Comment · Reviewer_s3ZC · 2021-09-07
> > **Response to Rebuttal**
> >
> > Thank you for your kind and detailed responses to my opinions. I think that this paper can be an important milestone for offline policy evaluation researches. Hence, I hold my score for the paper.

---

### Decision · Program_Chairs · 2021-09-27

**Decision:**

Accept (Poster)

**Comment:**

The manuscript examines the question of how to improve policy selection in the off-line RL setting. Typically offline policy selection is approached via off-policy evaluation (OPE), aimed at estimating the expected return of candidate policies. OPE is itself a difficult problem that typical requires hyperparameter tuning and selection itself. The paper develops moves closer to a hyperparameter-free method and demonstrates the effectiveness of the algorithm in the context of standardized offline datasets (e.g. RLUnplugged for Atari).
The algorithm for policy selection is built using insights from the recently published Batch Value-Function Tournament (BVFT) approach to estimating the best value function from among a set of candidates.
They make comparisons to well developed OPE style methods such as fitted Q-evaluation and show clear advantages in data efficiency and the policy selection.
The manuscript examines applying the approach to a wide range of settings (from Atari to continuous control) and to a range of policies produced by a variety of algorithms. The ideas, theory, and experiments are well motivated by the text. Taken together, the manuscript provides a promising look at a fundamental and open problem in RL.